# COLOR BLINDNESS TEST IMAGES AS SEEN BY LARGE VISION-LANGUAGE MODELS

## ABSTRACT

*Large vision-language models* (LVLMs) are fairly powerful in understanding this colorful world, yet their reasoning is grounded in *highly entangled semantics*, leaving open the question of whether they genuinely perceive colors in human-like manners. Although they could correctly answer questions related to colors, they might internally rely on specific *prior knowledge* and *correlations* between color and other semantics instead of directly process the color semantic. To this end, we study how LVLMs perceive *color blindness test images* (CBTIs), and we conclude that CBTIs as seen by LVLMs are different from CBTIs as seen by humans. Specifically, in this paper, we create *IshiharaColorBench* following the *Ishihara test*, where LVLMs have to directly process colors, and the digit in any test image could be recognized *if and only if* LVLMs genuinely perceive colors. We perform two types of tests: *standard color blindness tests* for performance assessment and *controlled color sensitivity tests* for behavior analysis. Given the former tests, LVLMs perform close to *random guessing*, and neither scaling-up nor fine-tuning leads to generalizable improvement; given the latter tests, we find several *systematic biases*, such as an imbalance in hue perception and the sensitivity to saturation contrast but not brightness contrast. Our findings reveal notable limitations of existing LVLMs in genuine color perception, thereby highlighting the need for developing novel model architectures or training strategies toward a smarter and more human-aligned perceptual foundation of LVLMs.

## 1 INTRODUCTION

The ability to perceive and interpret color is a fundamental faculty of visual intelligence, enabling both biological and artificial systems to navigate and comprehend their surroundings. For humans, this capacity represents a critical evolutionary advantage, essential for tasks ranging from identifying sustenance to recognizing danger (Jacobs, 2008). In modern artificial intelligence, this faculty is no less critical, underpinning the safety and reliability of applications ranging from autonomous driving, where correctly identifying the color of a traffic signal is non-negotiable, to medical image analysis, where subtle chromatic shifts can signify malignancy (Komura & Ishikawa, 2018). Consequently, endowing artificial intelligence with a robust, human-like understanding of color remains a central goal in computer vision (Gevers et al., 2012). The recent emergence of powerful large vision-language models (LVLMs) appears to mark a significant breakthrough, demonstrating impressive performance on a wide array of color-related tasks (Achiam et al., 2023; Bai et al., 2025; Wang et al., 2025; Liu et al., 2024d). However, their quantitative success on these benchmarks prompts a more profound, qualitative question: does this high performance truly equate to a genuine comprehension of color, a comprehension that can be trusted when lives and critical decisions are at stake?

Recent LVLMs, such as LLaVA (Liu et al., 2023), GPT-4V (Achiam et al., 2023), and Qwen2.5-VL (Bai et al., 2025), demonstrate impressive performance on standard color-related benchmarks. However, this proficiency becomes notably less reliable when the models are tested on inputs that defy common-sense expectations. When presented with a purple banana, a verdant sky, or a crimson-hued leaf in spring, the

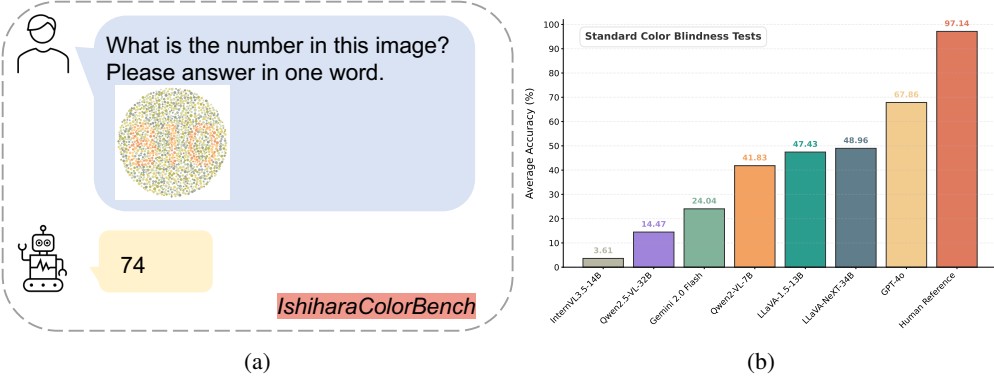

(a)             (b)

Figure 1: Performance of Large Vision-Language Models on the IshiharaColorBench. (a) An example from the benchmark, where the model is prompted to identify a number (810) defined purely by color. The test is designed to be illegible to individuals with common forms of color vision deficiency. (b) Performance comparison between various LVLMs and the human reference, revealing a profound performance gap.

models are prone to error, often disregarding the visual evidence to default to prototypical colors. This specific failure pattern is highly revealing: it suggests that the models' reasoning is not grounded in direct visual perception but is heavily mediated by learned priors and the highly entangled semantics of their training data (Leng et al., 2024; Liu et al., 2024c). This reveals a critical vulnerability: if a model's understanding of 'red' is merely an association with 'apple' or 'stop sign', it lacks the robust, first-principles perception needed for safe deployment in unpredictable, open-world environments. This reliance on priors is not an accidental flaw but a systemic artifact of how these models are evaluated (Lee et al., 2025; Lin et al., 2024). Existing benchmarks are almost exclusively composed of natural images where color is intrinsically entangled with object identity. Consequently, high scores on these benchmarks do not validate true color perception but rather a model's ability to recall semantic pairings, creating a dangerous illusion of competence.

Existing benchmarks (Liu et al., 2024d; Fu et al., 2024; Liang et al., 2025) present significant limitations for the robust evaluation of color perception. Their reliance on naturalistic images introduces a strong bias where object identity is frequently correlated with a prototypical color. Consequently, it is challenging to ascertain whether a model is demonstrating true color comprehension or is instead relying on learned object-color associations. A high accuracy score on such benchmarks may indicate a model's proficiency in recalling that bananas are yellow, rather than its ability to perceive the color yellow itself. This inherent confounding of semantic priors with visual perception means that current evaluation methods are insufficient for validating a generalized and unbiased color understanding. Therefore, a new evaluation paradigm is required—one that systematically isolates chromatic properties from object semantics to enable a more precise and faithful assessment of a model's perceptual capabilities.

To bridge this gap in evaluation benchmarks, we propose **IshiharaColorBench**, a pioneering benchmark engineered to disentangle chromatic processing from semantic influences (See Figure 1). Critically, IshiharaColorBench is more than a static dataset; it is an automated and configurable framework for generating diagnostic tests. It is comprised of two core components: *Standard Color Blindness Tests*, which mimic traditional Ishihara plates to assess overall performance, and *Controlled Color Sensitivity Tests*. This second component, powered by our procedural generation engine, provides fine-grained control over the HSV (Hue, Saturation, Value) color space (Wikipedia contributors, 2025a). This enables the creation of a virtually infinite set of customized stimuli to systematically probe models' perceptual boundaries, biases, and sensitivities with a precision previously unattainable. This framework ensures that success is possible if and only if LVLMs genuinely perceive color, forcing them to process raw pixel information instead of relying on semantic shortcuts.

Our evaluation of state-of-the-art LVLMs on this benchmark reveals a profound and consistent failure, a failure that directly questions their readiness for any color-critical application. On the *Standard Color Blindness Tests*, model performance is profoundly impaired; even top-performing models achieve accuracies that

are dramatically lower than the near-perfect human reference, falling well below any threshold for practical reliability (see Figure 1b). Crucially, we find that while scaling-up model size and extensive fine-tuning on our benchmark can lead to a certain degree of performance improvement, the models still fall significantly short of the human reference. This outcome strongly suggests that LVLMs' success on conventional benchmarks relies on entangled semantics rather than genuine color perception. Further, using the precision of our *Controlled Color Sensitivity Tests*, we uncover systematic biases, including an imbalanced perception across hues (e.g., a weakness on green tones) and a critical, non-humanlike reliance on saturation over brightness contrast. Such poor performance on green tones could cause a model's performance to degrade on specific tasks, such as crop recognition and traffic light judgment. Ultimately, IshiharaColorBench serves not just as a benchmark but as a crucial wake-up call, compelling the field to move beyond benchmarks that reward semantic shortcuts and toward building the truly robust perceptual foundation necessary for the next generation of safe and reliable AI.

In summary, our main contributions are:

- We introduce **IshiharaColorBench**, the first benchmark specifically designed to diagnose the genuine color perception of LVLMs by systematically disentangling it from semantic priors. Its core contribution is to challenge the prevailing illusion of competence created by existing benchmarks, establishing a necessary and rigorous standard for evaluating the foundational reliability of AI systems in color-critical, real-world applications.
- Through our benchmark's *Standard Color Blindness Tests*, we reveal a systemic and profound failure in state-of-the-art LVLMs. Their performance is profoundly impaired, falling dramatically short of the human reference and well below any threshold for reliable deployment. This finding provides a clear, quantified measure of the vast gap that remains between artificial and human-like color vision.
- We conduct the first in-depth analysis to deconstruct how LVLMs fail at color perception through our novel *Controlled Color Sensitivity Tests*. These tests, systematically generated with precise procedural control over individual color attributes (HSV), allow us to pinpoint specific, systematic biases for the first time. Our analysis reveals a non-humanlike reliance on saturation over brightness and a non-uniform sensitivity across hues, providing a crucial diagnostic map of the models' internal mechanisms that can guide future research toward building truly robust and human-aligned perceptual systems.

## 2 RELATED WORK

**The Rise of Large Vision-Language Models:** The landscape of artificial intelligence has been fundamentally reshaped by the emergence of Large Vision-Language Models (LVLMs). These architectures achieve a powerful synthesis by coupling pre-trained vision encoders, such as Vision Transformers (ViTs) (Vaswani et al., 2017), with large language models (LLMs) (Grattafiori et al., 2024; Liu et al., 2024a; Achiam et al., 2023). Early pioneering work in multimodal learning, like CLIP (Radford et al., 2021), demonstrated the immense potential of learning joint representations from web-scale image-text pairs, enabling remarkable zero-shot classification capabilities. Building on this foundation, a new wave of instruction-following models has been developed. This includes powerful proprietary systems like GPT-4V (Achiam et al., 2023), as well as a vibrant ecosystem of open-source models such as LLaVA (Liu et al., 2023), Qwen2.5-VL (Bai et al., 2025), and InternVL (Wang et al., 2025). These models exhibit stunning abilities in complex, high-level tasks such as detailed image captioning, visual question answering (VQA), and nuanced human-AI dialogue (Liu et al., 2024d). However, this rapid progress in generative and conversational skills creates a paradox: while models can eloquently describe visual scenes, the depth and robustness of their underlying perceptual grounding remain surprisingly under-examined. Our research addresses this gap by shifting the focus from high-level reasoning to the rigorous assessment of a fundamental perceptual faculty: color.

**Challenges in Evaluating LVLMs:** As LVLMs become more capable, the research community recognizes that standard benchmarks measuring aggregate performance are insufficient for true model understanding. This has spurred a necessary shift towards diagnostic evaluation—creating targeted probes to assess specific,

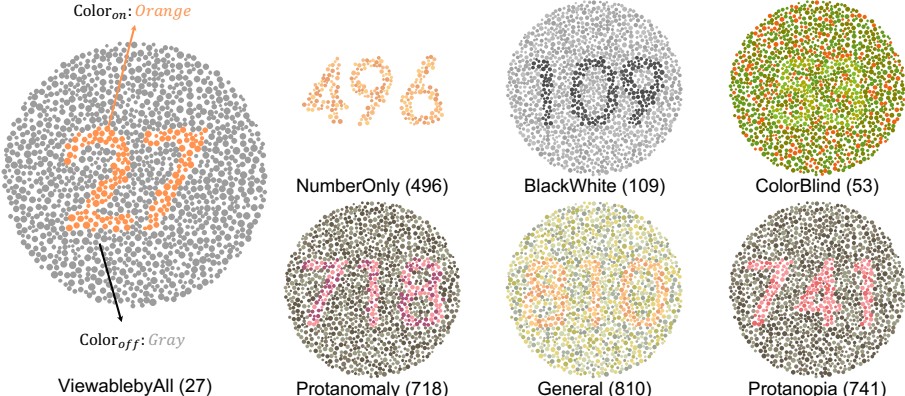

Figure 2: Samples in Standard Color Blindness Tests. We also provide simulated images as they would be seen by individuals with different types of color blindness, as shown in Appendix.

foundational skills (Liu et al., 2024d). A significant body of work has emerged to test discrete abilities, including spatial relationship understanding (Wang et al., 2024a), numeracy and object counting (Fu et al., 2024), and the propensity for object hallucination where models invent objects not present in the image (Li et al., 2023; Wang et al., 2023). A critical, recurring theme in these diagnostic efforts is the challenge of disentanglement. For instance, when evaluating color perception, existing benchmarks commonly rely on semantically-rich queries, such as asking for the color of a familiar object (e.g., "What color is the stop sign?") (Liang et al., 2025). This evaluation paradigm is inherently ambiguous. It fails to distinguish between genuine, real-time color perception and the retrieval of stored semantic knowledge (i.e., the fact that stop signs are usually red). This confounding factor makes it impossible to ascertain the model's true perceptual abilities, motivating the critical need for an evaluation methodology that can effectively isolate a perceptual skill from the vast web of semantic associations the model has learned.

**The Ishihara Test: A Medically Validated Paradigm for Color Perception:** The Ishihara test, developed by Dr. Shinobu Ishihara in 1917, is a highly reliable method for diagnosing red-green color vision deficiency (CVD) (Ishihara, 1917). Its success stems from its "pseudoisochromatic plates", which are designed to isolate color as the sole discriminating feature. These plates consist of a mosaic of dots where a figure is distinguished from the background only by hue. To ensure the test specifically assesses color perception, other visual cues are neutralized. The dots for both the figure and background are intentionally varied in size and luminance, preventing observers from using brightness or texture to identify the figure (Birch, 1997). Validated over a century of clinical use, the Ishihara test is the gold standard for CVD screening, consistently demonstrating high sensitivity and specificity. Studies have confirmed its ability to identify red-green deficiencies with up to 97% sensitivity and 100% specificity (Cole et al., 2006). This establishes the Ishihara plates as a scientifically validated paradigm for assessing a specific perceptual skill in isolation.

## 3 A CLOSER LOOK AT THE COLORBLINDNESS OF LVLMS

### 3.1 STANDARD COLOR BLINDNESS TESTS

The visibility metrics in this study are derived from the functional design of the Ishihara Color Vision Test, a long-established benchmark for identifying Red-Green Color Deficiency (RCD) (Wald & Brown, 1965). All test data are created based on the principle of pseudoisochromaticity. This design methodology ensures that patterns share similar levels of perceived brightness (luminance) but differ in their color properties (chrominance), particularly along the red-green axis (Wong, 2011). To systematically evaluate performance, we organize our data into three groups. The first, "Universal Visibility", includes patterns designed with brightness contrast to be perceivable by all observers, regardless of their color vision capabilities. The second, "Preferential Visibility for Normal Vision", contains data where information is encoded in the red-

Table 1: Model Evaluation on Standard Color Blindness Tests in IshiharaColorBench. All values are percentages (%).

| | Universal Visibility (Visible to All Individuals) | | | Preferential Visibility for Normal Vision | | | Preferential Visibility for RCD |
|---|---|---|---|---|---|---|---|
| | ViewablebyAll | BlackWhite | NumberOnly | General | Protanomaly | Protanopia | Colorblind |
| *Human Reference* | 100.00 | 100.00 | 100.00 | 100.00 | 100.00 | 100.00 | 80.00 |
| *Closed-source Models* | | | | | | | |
| Gemini 2.0 Flash | 14.50 | 49.30 | 99.80 | 1.10 | 1.50 | 2.10 | 0.10 |
| GPT-4o | 99.20 | 70.90 | 99.70 | 54.80 | 97.10 | 93.20 | 0.10 |
| *Open-source Models* | | | | | | | |
| LLaVA-NeXT-34B | 85.20 | 59.10 | 94.40 | 9.90 | 52.00 | 71.80 | 0.30 |
| Qwen2.5-VL-32B | 4.00 | 1.70 | 94.40 | 0.10 | 0.30 | 0.30 | 0.10 |
| InternVL3.5-14B | 2.90 | 0.80 | 19.90 | 0.20 | 0.10 | 1.00 | 0.10 |
| LLaVA-NeXT-13B | 64.80 | 41.80 | 89.70 | 8.10 | 32.10 | 45.80 | 0.10 |
| LLaVA-1.5-13B | 68.80 | 49.20 | 82.10 | 16.60 | 48.70 | 58.90 | 0.10 |
| InternVL3.5-8B | 0.30 | 0.30 | 17.70 | 0.10 | 0.40 | 0.30 | 0.10 |
| Qwen2.5-VL-7B | 0.90 | 1.20 | 73.20 | 0.00 | 0.40 | 0.20 | 0.10 |
| Qwen2-VL-7B | 48.30 | 25.70 | 99.80 | 1.10 | 8.80 | 13.60 | 0.10 |
| LLaVA-NeXT-7B | 56.30 | 29.50 | 86.30 | 3.50 | 23.20 | 40.10 | 0.10 |
| LLaVA-1.5-7B | 66.40 | 41.30 | 74.20 | 10.40 | 36.20 | 55.00 | 0.10 |
| InternVL3.5-4B | 1.00 | 1.40 | 5.60 | 0.10 | 0.30 | 0.30 | 0.00 |
| Qwen2.5-VL-3B | 0.10 | 0.10 | 30.80 | 0.10 | 0.00 | 0.20 | 0.00 |
| InternVL3.5-2B | 0.10 | 0.30 | 37.30 | 0.10 | 0.00 | 0.00 | 0.00 |
| Qwen2-VL-2B | 38.50 | 5.00 | 99.80 | 0.20 | 2.70 | 5.70 | 0.00 |
| InternVL3.5-1B | 0.70 | 0.30 | 14.50 | 0.40 | 0.50 | 0.50 | 0.10 |

green color channels, making it easily readable for individuals with typical vision but ambiguous or misread by those with RCD. The third group, "Preferential Visibility for RCD", features a unique category that mimics the "hidden-digit" plates of the Ishihara test, where the embedded figure is paradoxically more discernible to individuals with RCD than to those with normal vision. All types are shown in Figure 2. For a detailed explanation of these data groups and their subtypes, please refer to Appendix A.1.

## 3.2 ANALYSIS OF MODEL PERFORMANCE ON THE STANDARD COLOR BLINDNESS TESTS

**Experimental Setup.** To systematically evaluate the color perception capabilities of modern LVLMs, we assess their performance on our comprehensive IshiharaColorBench. This benchmark contains 7,000 images distributed across seven sub-categories, with each category representing digits from 0 to 999. It is crucial to frame the underlying challenge correctly: at its core, this is a straightforward digit recognition task where the primary obstacle is purely perceptual. Therefore, any visually competent system should achieve near-perfect accuracy (e.g., $> 95\%$), making any significant deviation a sign of a fundamental perceptual deficiency. Our evaluation spans a wide array of prominent models, including closed-source systems (Gemini 2.0 Flash (Comanici et al., 2025), GPT-4o (Achiam et al., 2023)) and major open-source models (LLaVA-1.5 (Liu et al., 2023), LLaVA-Next (Liu et al., 2024b), Qwen2-VL (Wang et al., 2024b), Qwen2.5-VL (Bai et al., 2025) and InternVL3.5 (Wang et al., 2025)). To ground these results against a practical baseline, we also established a *Human Reference* score. This score is derived from a preliminary evaluation conducted by a group of non-colorblind human participants, consisting of the authors and our peers, and serves to approximate human-level perception on this task. The results in Table 1 form the basis for the following critical findings.

**A Widespread Failure on a Foundational Perceptual Task.** Table 1 reveals a systemic failure of current LVLMs on a task that is trivial for human perception. The vast majority of models are effectively blind to the figures in the *'ViewablebyAll'* plates, which are designed to be universally visible. Most models, such as Gemini 2.0 Flash at 14.50%, Qwen2.5-VL-32B at 4.00%, and InternVL3.5-14B at a mere 2.90%, perform drastically below any threshold of competence. Their incorrect responses are often not even reasonable guesses, but rather confident assertions that no number is present, highlighting a profound lack of perceptual awareness. This inability to process patterns defined purely by chromatic differences—without strong shape

or brightness cues—is a classic sign of a system that lacks true color processing. Even advanced models like GPT-4V have shown significant limitations when subjected to Ishihara-style tests, underscoring the persistent challenge these assessments pose for computer vision. This starkly contrasts with the 100% human accuracy, revealing a critical gap between the heralded semantic reasoning of modern AI and its foundational visual perception, and questioning their reliability in any scenario where subtle visual cues are paramount.

**Inconsistent Scaling Laws and the Primacy of Architecture.** The data presents a direct challenge to the "bigger is better" narrative, showing that scaling laws are not a universal guarantee of improved perception. While performance consistently scales *within* a given architectural family—for instance, LLaVA-NeXT improves from 56.30% at 7B to 85.20% at 34B—this trend completely breaks down when comparing *across* different families. The 34B LLaVA-NeXT, for example, massively outperforms the similarly-sized Qwen2.5-VL-32B (4.00%). More strikingly, the smaller LLaVA-1.5-13B model (68.80%) is an order of magnitude better than the larger InternVL3.5-14B (2.90%). Remarkably, even the 7B-parameter LLaVA-1.5 model at 66.40% is vastly superior to the 32B Qwen2.5-VL, showcasing a multi-billion parameter gap in perceptual efficiency. Research indicates that architectural choices can significantly influence scaling behavior, with some models showing poor scalability despite comparable performance at smaller sizes. This is a crucial finding for the field, as it suggests that design choices made early in a model's development, such as the composition of its visual pre-training data or the specific architecture of its vision encoder, create a lasting legacy on its perceptual capabilities that cannot be easily papered over with more parameters.

**A Rigid "Normal Vision" Bias Without Perceptual Flexibility.** The performance patterns across visibility categories show that LVLMs have internalized a rigid perceptual system biased toward typical human vision. GPT-4o, the best-performing model overall, exemplifies this by excelling on plates designed for normal vision, scoring 97.10% on *'Protanomaly'* and 93.20% on *'Protanopia'* tests. However, its performance plummets to a near-zero 0.10% on the *'Colorblind'* test, where the hidden figure is visible to those with red-green color deficiency. In stark contrast, our non-colorblind human evaluators achieved 80.00% accuracy on this same counter-intuitive task. This human success is attributed to the ability to override initial perceptual biases and apply focused reasoning to detect subtle, unconventional patterns. Studies show that while LVLMs may possess factual knowledge about color vision deficiencies, they fail to simulate how individuals with these conditions perceive color in image-based tasks. The models' complete failure indicates that their internal color representation is not a principled, continuous space, but likely a collection of discrete, learned associations that lack the robustness to handle out-of-distribution chromatic arrangements.

**An Alarming Trend of Performance Regression in Newer Models.** Perhaps the most concerning discovery is a clear trend of performance degradation in newer, supposedly superior model generations, suggesting that the pursuit of general capabilities may inadvertently erode foundational skills. The evidence is unambiguous: the older Qwen2-VL-7B (48.30%) is over 50 times more effective on the *'ViewablebyAll'* task than its successor, Qwen2.5-VL-7B (0.90%). A similar, consistent regression is observed in the LLaVA family, where the older LLaVA-1.5 models outperform their newer LLaVA-NeXT counterparts at both the 13B and 7B scales. We hypothesize this is an unintended consequence of recent training priorities; as developers fine-tune for complex reasoning and OCR on high-contrast documents, they may inadvertently de-prioritize or even damage the model's sensitivity to subtle chromatic variations. This phenomenon represents a critical perceptual blind spot in current AI development practices, where progress on one axis can mask severe degradation on another, more fundamental one.

**A Generalization Failure Rooted in Visual Pre-training.** We note that the *'NumberOnly'* category involves minimal color distraction relative to other tasks. Nevertheless, many models still fail to achieve 90% accuracy on this simple digit recognition task. This led us to question whether the poor performance on our benchmark stems from a fundamental lack of capacity. To diagnose the root cause of these failures, further experiments (detailed in Appendix A.3.1) confirm the issue is one of generalization, not an inherent architectural incapacity. Models can be fine-tuned to near-perfect accuracy on our benchmark, proving they possess the necessary capacity, but this knowledge is brittle and fails to transfer. For instance, a fine-tuned

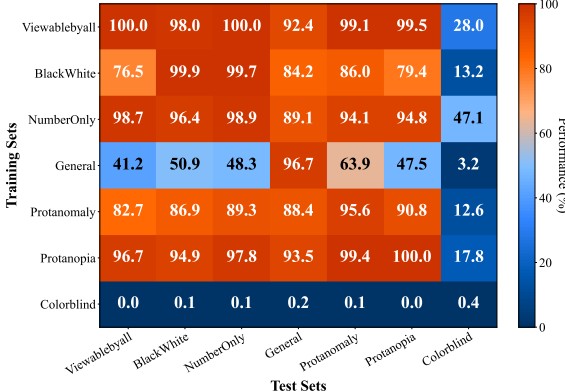

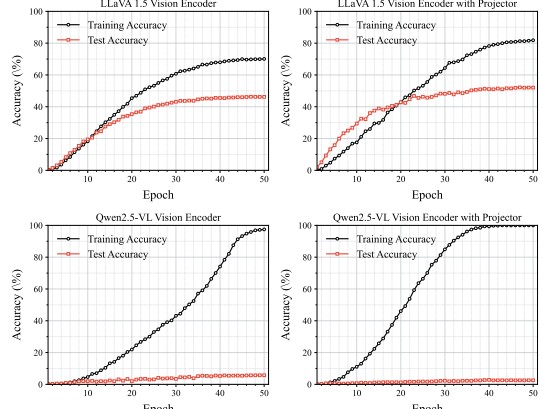

Figure 3: Performance comparison of LLaVA-1.5-7B model after LoRA fine-tuning. Columns represent different test sets (the model is evaluated on these datasets), and rows represent different LoRA training sets. All LoRA training parameters are kept consistent to ensure the fairness of performance comparison. The color depth and numerical value indicate the model's performance on the corresponding test set, with higher values representing better performance.

Figure 4: Linear probing results of vision encoders with/without projector. Four subplots correspond to LLaVA 1.5/Qwen2.5-VL encoders (with/without frozen projector). A 1-layer FC classifier was attached to frozen components, trained on a data subset and tested on another. Curves show training/test accuracy across epochs, comparing projector's impact on performance.

model essentially memorizes the specific color pairings in the training set rather than learning the abstract concept of 'figure-ground segregation via color contrast'. This indicates they learn isolated "tricks" rather than a robust visual principle. Linear probing analyses further pinpoint the bottleneck to the visual encoder, implying that the visual features required for this task are either never learned or are actively suppressed during pre-training in favor of features more useful for object recognition, such as shape and texture. This provides a clear direction for future work and motivates our subsequent investigation into the specific perceptual dimensions that models struggle with.

## 4 PERCEPTION OF COLOR-ONLY SEMANTICS IN LVLMS

### 4.1 CONTROLLED COLOR SENSITIVITY TESTS

While the *Standard Color Blindness Tests* effectively reveal that LVLMs fail, they do not fully explain how they fail. Their plates often confound multiple color properties (e.g., simultaneous changes in hue, saturation, and brightness), making it difficult to isolate the precise causes of perceptual deficits. To deconstruct these failures, we introduce a core contribution of our work: an automated procedural generation framework for creating pseudo-isochromatic test images with granular, independent control over their color attributes.

This powerful framework is the engine behind our controlled experiments. It programmatically generates Ishihara-style plates by operating within the Hue-Saturation-Value (HSV) color space (Wikipedia contributors, 2025a), which allows for the intuitive and orthogonal manipulation of core color components. By specifying distinct HSV values for the foreground figure and the background, our tool can automate the creation of a virtually limitless suite of diagnostic stimuli. This enables a series of controlled experiments where one chromatic dimension (e.g., Saturation) is systematically varied while the others are held constant, allowing us to precisely measure a model's sensitivity to isolated changes. The specific algorithms for pointillistic rendering and color selection are detailed in Appendix B.1.

This methodology forms the foundation for the key analyses presented in the following sections, where we systematically probe for specific perceptual biases. While we discuss the main findings in the body of the

paper, the full experimental protocol, additional results, and detailed analysis for each sensitivity test are provided in Appendix B.3. Ultimately, this approach allows us to move beyond simply asking if models perceive color, to rigorously investigating how they perceive its individual components.

### 4.2 ANALYSIS OF CONTROLLED COLOR SENSITIVITY ACROSS H, S, AND V DIMENSIONS

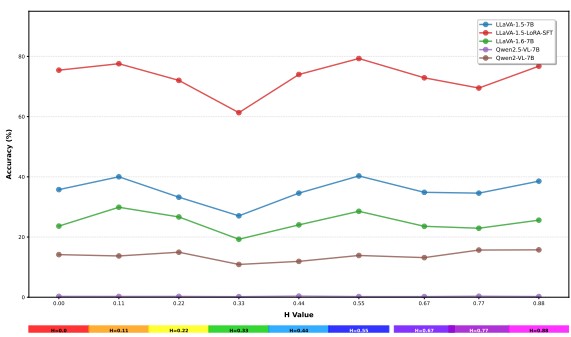

Figure 5: Model accuracy across H values. Each data point represents the average of ten trials with varying Saturation and Value.

**Systemic Color Perception Imbalance with a Vulnerability in Green-Yellow Hues.** Our analysis of model performance across the color spectrum reveals a significant and systemic imbalance, with a shared vulnerability to hues in the green-yellow range. This conclusion is strongly supported by the trend illustrated in Figure 5, which displays the average accuracy from ten distinct runs where Hue (H) was fixed while Saturation (S) and Value (V) were varied. As the figure demonstrates, most evaluated models, including both LLaVA and Qwen variants, experience a sharp drop in accuracy around H values of 0.22 and 0.33. While performing well in blue and red regions, their accuracy plummets when identifying shades of green, suggesting a systemic bias in how they process and interpret certain color ranges. This perceptual gap is not a minor flaw; it has critical implications for real-world applications where precise color identification is paramount, such as recognizing traffic lights for autonomous vehicles or assessing crop health in agriculture. The consistency of this failure across different model families points to a fundamental issue, likely rooted in training data distribution or inherent feature extraction biases, highlighting the need for new benchmarks and targeted data augmentation to ensure uniform color perception.

**A Profound Asymmetry in Saturation Perception Unlike Human Vision.** Controlled tests in the Saturation (S) dimension, detailed in Table 2, reveal a critical departure from human-like perception. Models exhibit a profound directional bias, showing they are not invariant to foreground-background color swaps in the way a human observer is. This is starkly illustrated by the Qwen2-VL-7B model, which achieves 86.63% accuracy on a high-saturation figure against a medium-saturation background ($S_{\text{On}} = 1$, $S_{\text{Off}} = 0.3$) but catastrophically fails at 0.43% when the roles are reversed ($S_{\text{On}} = 0.3$, $S_{\text{Off}} = 1$). This vast performance gulf demonstrates that the model is not performing robust figure-ground separation but is instead reliant on detecting a simple high-purity "pop-out" signal. The failures in these Saturation tests are consistently more severe than those in the Value (brightness) dimension, indicating that variations in color purity represent a more critical weakness for current LVLMs.

**Significant Robustness Gaps Between Model Families in Brightness Perception.** Our analysis of the Value (V) dimension, reveals clear and consistent performance divides between different model families in their ability to handle contrast variations. As detailed in Table 2, the LLaVA family demonstrates significantly greater robustness in challenging low-contrast scenarios. For instance, LLaVA 1.5-13B maintains 48.50% accuracy on a bright numeral against a very dark background ($V_{\text{On}} = 1$, $V_{\text{Off}} = 0.1$), while the entire Qwen2.5-VL family scores near-zero (e.g., 0.17% for the 32B model) under the same conditions. This stark divergence suggests that the architectural or training strategies of the LLaVA family are more effective at preserving the ability to process brightness variations. While the directional bias in the Value dimension is less extreme than the asymmetries seen in Saturation, the complete failure of an entire model family under specific conditions underscores a critical and addressable weakness in perceptual training.

Table 2: Controlled Color Sensitivity Tests: Model Performance Across Saturation (S) and Value (V) Dimensions. The statistics are averaged across the Hue (H) dimension. All values are percentages (%).

| S | | V | | Qwen2.5-VL | | | Qwen2-VL | | LLaVA 1.5 | | LLaVA Next | |
|---|---|---|---|---|---|---|---|---|---|---|---|---|
| On | Off | On | Off | 32B | 7B | 3B | 7B | 2B | 13B | 7B | 13B | 7B |
| 1 | 1 | 1 | 0.7 | 0.03±0.06 | 0.10±0.10 | 0.03±0.06 | 0.10±0.10 | 0.07±0.06 | 0.20±0.26 | 0.13±0.15 | 0.13±0.06 | 0.20±0.10 |
| 1 | 1 | 1 | 0.5 | 0.07±0.06 | 0.07±0.06 | 0.03±0.06 | 0.53±0.06 | 0.67±0.42 | 5.30±5.15 | 4.93±3.01 | 2.53±2.91 | 1.37±1.15 |
| 1 | 1 | 1 | 0.3 | 0.17±0.06 | 0.13±0.06 | 0.13±0.15 | 2.83±1.93 | 1.93±1.25 | 13.73±19.25 | 19.50±18.15 | 13.03±16.57 | 12.70±11.50 |
| 1 | 1 | 1 | 0.1 | 0.17±0.15 | 0.17±0.15 | 0.10±0.00 | 5.30±4.95 | 2.67±1.92 | 48.50±9.12 | 49.73±14.15 | 39.97±14.41 | 27.77±11.25 |
| 1 | 1 | 0.7 | 1 | 0.03±0.06 | 0.17±0.06 | 0.07±0.06 | 0.17±0.06 | 0.17±0.06 | 0.20±0.10 | 0.10±0.00 | 0.10±0.00 | 0.10±0.00 |
| 1 | 1 | 0.5 | 1 | 0.13±0.06 | 0.07±0.06 | 0.07±0.06 | 0.43±0.06 | 0.27±0.12 | 4.70±1.65 | 3.50±1.48 | 2.47±1.01 | 0.93±0.15 |
| 1 | 1 | 0.3 | 1 | 0.07±0.06 | 0.10±0.00 | 0.00±0.00 | 1.80±1.51 | 0.53±0.25 | 27.13±13.50 | 28.67±16.85 | 25.20±11.85 | 13.23±9.95 |
| 1 | 1 | 0.1 | 1 | 0.10±0.00 | 0.27±0.06 | 0.07±0.06 | 10.30±12.65 | 1.73±1.55 | 50.20±7.25 | 50.03±5.15 | 37.47±2.91 | 25.03±7.15 |
| 1 | 0.7 | 1 | 1 | 0.17±0.15 | 0.20±0.10 | 0.00±0.00 | 0.70±0.10 | 0.33±0.25 | 0.97±0.76 | 0.77±0.55 | 0.37±0.25 | 0.27±0.15 |
| 1 | 0.5 | 1 | 1 | 0.27±0.06 | 0.20±0.10 | 0.10±0.10 | 18.63±2.25 | 4.37±2.37 | 33.50±10.45 | 20.40±11.35 | 19.93±11.41 | 11.93±5.85 |
| 1 | 0.3 | 1 | 1 | 5.60±3.21 | 1.93±0.45 | 0.73±0.55 | 86.63±4.95 | 75.90±12.75 | 68.30±3.15 | 63.13±3.25 | 61.27±5.35 | 64.93±4.75 |
| 1 | 0.1 | 1 | 1 | 45.90±3.06 | 23.43±2.25 | 12.33±3.45 | 98.33±0.25 | 98.17±0.25 | 78.03±0.55 | 77.47±2.25 | 80.20±2.45 | 80.07±1.65 |
| 0.7 | 1 | 1 | 1 | 0.03±0.06 | 0.10±0.00 | 0.00±0.00 | 0.13±0.06 | 0.10±0.10 | 0.23±0.15 | 0.20±0.10 | 0.10±0.00 | 0.13±0.06 |
| 0.5 | 1 | 1 | 1 | 0.13±0.06 | 0.13±0.06 | 0.17±0.06 | 0.13±0.06 | 0.20±0.10 | 6.27±3.55 | 6.47±2.95 | 3.50±2.35 | 2.77±1.35 |
| 0.3 | 1 | 1 | 1 | 0.10±0.10 | 0.03±0.06 | 0.00±0.00 | 0.43±0.25 | 0.13±0.06 | 39.70±10.25 | 38.47±9.25 | 31.07±5.15 | 34.57±4.05 |
| 0.1 | 1 | 1 | 1 | 0.13±0.06 | 0.17±0.15 | 0.10±0.00 | 4.03±0.55 | 0.43±0.15 | 56.30±3.55 | 54.70±3.75 | 44.20±2.55 | 50.00±5.65 |

**Fragmented Color Learning and a Failure to Generalize Across Hues.** To probe the nature of this generalization failure, we conducted targeted LoRA fine-tuning experiments, training the model on specific color channels (e.g., Red) and testing its ability to perceive others (e.g., Green, Blue). The results reveal that while the model can effectively learn to master color discrimination for a specific hue it is trained on, this learning is highly fragmented and fails to transfer to other colors. For instance, a model fine-tuned to expertly identify red figures remains effectively "blind" to green or blue ones, even under identical high-contrast conditions. This demonstrates that the model is not learning a universal, hue-agnostic principle of color contrast; instead, it develops isolated, non-transferable skills for each color dimension. This fragmented perception provides a deeper explanation for the widespread failures observed in our benchmark, as the models lack a unified and robust understanding of the color space. A full breakdown of these cross-channel generalization experiments, including detailed results and visualizations, is available in the Appendix B.4.

## 5 CONCLUSIONS AND FUTURE DISCUSSIONS

In this paper, we confront the challenge of evaluating genuine color perception in LVLM by introducing *IshiharaColorBench*. This benchmark, powered by a procedural generation framework, is specifically designed to isolate chromatic processing from the confounding influence of semantic priors, moving beyond the limitations of existing, entanglement-prone evaluations.

Our comprehensive experiments reveal a profound deficiency in the current generation of SOTA LVLMs. On our *Standard Color Blindness Tests*, model performance is profoundly impaired, falling dramatically short of the human baseline. This is not a problem of scale or knowledge; we show that neither larger models nor task-specific fine-tuning fosters a generalizable understanding of color, leading instead to mere overfitting. Furthermore, our *Controlled Color Sensitivity Tests* illuminate the why behind this failure, uncovering systematic biases that are fundamentally non-humanlike: an imbalanced perception across hues (e.g., green tones) and a critical over-reliance on saturation contrast at the expense of brightness. These findings provide conclusive evidence that current LVLMs do not reason from first-principles visual perception; instead, they employ a brittle strategy of recalling learned semantic associations.

The implications of our work extend beyond the domain of color. They serve as a call for a paradigm shift in how we build and evaluate perceptual abilities in AI. Future research must venture beyond simply scaling up entangled datasets and instead explore novel pre-training objectives that enforce perceptual consistency or investigate architectures with explicit mechanisms for disentangled feature representation. Ultimately, *IshiharaColorBench* is more than a diagnostic tool; it is a necessary step toward building the next generation of AI systems—those that not only predict but genuinely perceive the world, forming the bedrock of truly reliable and trustworthy visual intelligence.

## USAGE OF LARGE LANGUAGE MODELS

In this paper, we adapt some large language models, such as ChatGPT 5 and Gemini 2.5, solely to assist with language refinement and polishing of the manuscript. They are not used for generating research ideas. designing methods, or conducting literature retrieval and discovery.

## ETHICS STATEMENT

This research was conducted with a commitment to ethical principles and responsible scientific practice. Our study involves two main components with ethical considerations: the evaluation of human participants and the broader impact of our AI benchmark.

**Human Evaluation.** The human performance data used as a reference in our study was collected from a small group of volunteers, including the authors and fellow students from the authors' institution. All participants provided informed consent prior to the study and were aware that their anonymized results would be used for academic research purposes. The participants all self-reported having normal color vision. The task, which involved identifying numbers in Ishihara-style plates, is a standard, non-invasive procedure that poses no risk to participants' well-being. All collected data was fully anonymized, and we only report aggregated statistics (e.g., average accuracy) to ensure the privacy of all individuals.

**AI Benchmark and Societal Impact.** The primary goal of IshiharaColorBench is to improve the safety and reliability of large vision-language models. By rigorously diagnosing a fundamental flaw in their perceptual abilities, our work contributes to a more realistic understanding of current AI capabilities and encourages the development of more robust systems. We believe this is a crucial step toward preventing model failures in safety-critical, real-world applications. Furthermore, our benchmark's data is procedurally generated with precise mathematical control over the color space. This methodology ensures that the evaluation is objective and free from the social biases that are often embedded in large-scale, web-scraped datasets. We do not foresee any significant negative dual-use applications for this research; on the contrary, we hope it serves as a valuable diagnostic tool for the entire AI research community to build better and safer models.

## REPRODUCIBILITY STATEMENT

To ensure the full reproducibility of our findings, we commit to making all necessary artifacts publicly available upon publication. Our reproducibility efforts are centered on three key components: the data generation framework, the experimental evaluation protocol, and the analysis code.

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

# A  STANDARD COLOR BLINDNESS TESTS

## A.1  DATA CONSTRUCTION

The visibility metrics in this study are derived from the functional design of the Ishihara Color Vision Test, a long-established benchmark for identifying red-green color deficiency (RCD) . All test data are created based on the principle of *pseudoisochromaticity*. This design methodology ensures that patterns share similar levels of perceived brightness (luminance) but differ in their color properties (chrominance), particularly along the red-green axis . This intentional design causes the patterns to be perceived differently by individuals with typical color vision versus those with RCD.

Our data are organized into three groups, as shown in Figure 2.

**Universal Visibility**   This group includes patterns designed to be perceivable by all observers, as their visibility depends on brightness contrast rather than color differentiation. This group contains three types:

- *ViewablebyAll*: Uses a high-contrast bright orange foreground against a neutral gray background. The stark difference in brightness makes it clearly visible to all observers.

- *BlackWhite*: Employs a grayscale palette as a baseline control that relies solely on luminance. By removing all color information, it provides a pure baseline for visibility assessment.

- *NumberOnly*: Presents a figure in shades of orange against a plain white background to maximize the brightness difference, ensuring easy readability.

**Preferential Visibility for Normal Vision**   This group contains data that are easily readable by individuals with typical trichromatic vision but are ambiguous or misread by those with RCD. This effect is achieved by encoding the primary information in the red-green color channels.

- *General*: Places an orange figure against a grayish-green background (Wikipedia contributors, 2025b). For those with typical vision, the colors are distinct, but individuals with RCD perceive them as similar hues, making the figure difficult to discern.

- *Protanomaly*: Targets red-weakness by using a magenta and pink foreground against a dark, earthy background (Wikipedia contributors, 2025b). Individuals with protanomaly have reduced sensitivity to red light, causing the foreground colors to appear muted and blend in with the background.

- *Protanopia*: Tests for a complete lack of red-sensing photoreceptors using a red and pink foreground. Individuals with this condition perceive these colors as dark, desaturated tones, effectively hiding the figure against the dark background (Wikipedia contributors, 2025b).

**Preferential Visibility for RCD**   This group features a unique category where the embedded figure is more discernible to individuals with RCD than to those with normal vision.

- *Colorblind*: Mimics the "hidden-digit" plates of the Ishihara test (Wikipedia contributors, 2025b). The figure is rendered in yellow-green hues against a complex background of olive greens and a disruptive orange. For an observer with typical vision, the similarity in hues and complex texture makes the figure difficult to separate. Conversely, for some individuals with RCD, their visual system's processing of these specific colors can introduce perceptual artifacts that create a discernible boundary, revealing the figure.

## A.2 Examples from Standard Color Blindness Tests

An analysis of the simulated images (See Figure 6 - 9) reveals distinct changes in the discriminability of data belonging to different categories. For simulations of both Deutan (green-deficient) and Protan (red-deficient) vision, including both *weak* and *blind* conditions, the distinguishability of data in the "General", "Protanomaly", and "Protanopia" categories is significantly diminished. The color cues that separate these data points from their background are lost, making them difficult to identify.

A similar trend is observed in the Tritan (blue-deficient) simulations. In both the *weak* and *blind* states, data points corresponding to the "General" category become less discernible. The simulations demonstrate that the loss of blue-yellow color information specifically impacts the visibility of this category.

Conversely, a consistent and noteworthy improvement is seen across all dichromatic and anomalous trichromatic simulations. The discriminability of data in the "Colorblind" category is markedly increased in every Deutan, Protan, and Tritan simulation. The removal of specific color channels appears to enhance the luminance contrast of this data, making it stand out more clearly.

Finally, under the Monochromacy simulations, which represent a complete or near-complete lack of color vision, the effect is most severe. Data points for the "General", "Protanomaly", and "Protanopia" categories become almost entirely indistinguishable from their surroundings. Yet, even in this grayscale environment, the "Colorblind" category data remains highly conspicuous and easily identifiable due to its inherent contrast.

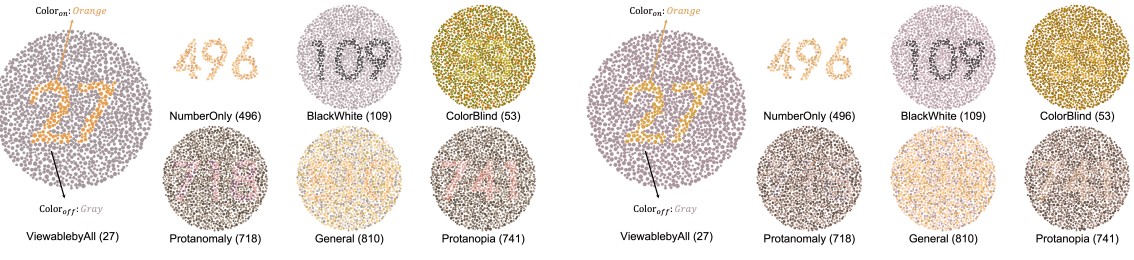

(a) Green-Weak/Deuteranomaly             (b) Green-Blind/Deuteranopia

Figure 6: Deutan Color Vision Deficiency Simulations

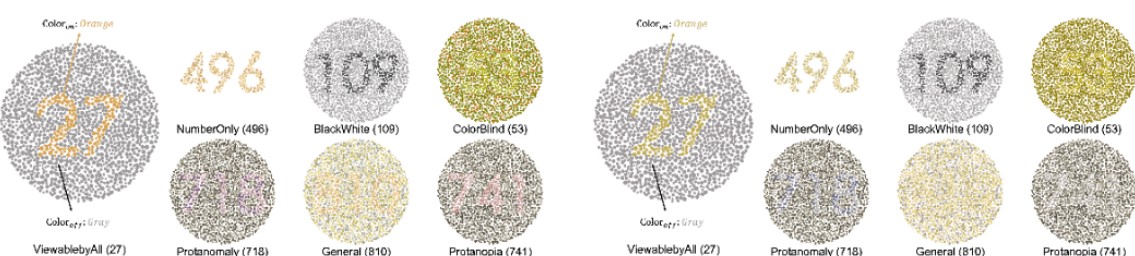

(a) Red-Weak/Protanomaly             (b) Red-Blind/Protanopia

Figure 7: Protan Color Vision Deficiency Simulations

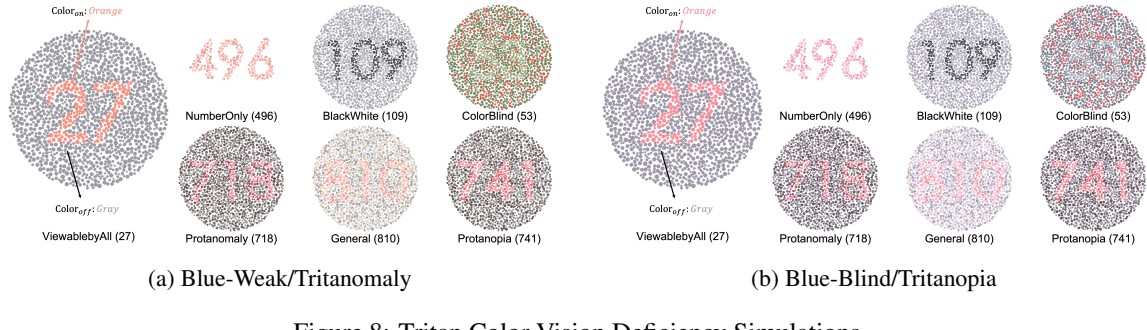

(a) Blue-Weak/Tritanomaly        (b) Blue-Blind/Tritanopia

Figure 8: Tritan Color Vision Deficiency Simulations

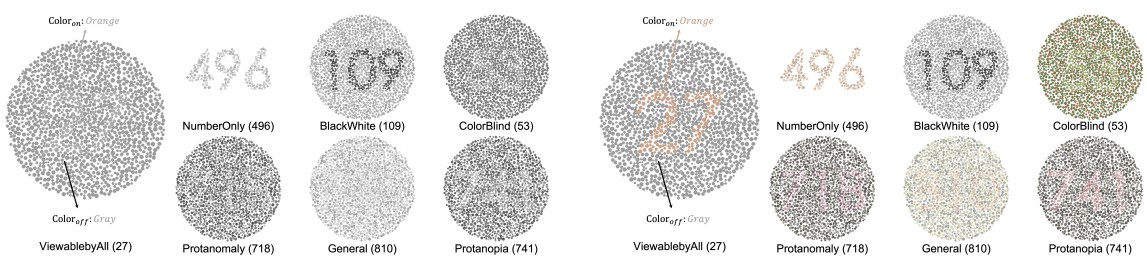

(a) Monochromacy/Achromatopsia        (b) Blue Cone Monochromacy

Figure 9: Monochromacy (Black & White) Vision Simulations

## A.3 DETAILED EXPERIMENT RESULTS AND ANALYSIS

### A.3.1 UNLOCKING LATENT ABILITIES: A DIAGNOSTIC ANALYSIS VIA FINE-TUNING AND PROBING

Given the widespread failure of pre-trained LVLMs, a critical question arises: is this an inherent limitation of their architecture, or simply a result of data exposure? To dissect this problem, we conducted controlled experiments to investigate the model's capacity to learn this task. It is important to remember that this is a simple digit recognition task where high accuracy, well above 90%, would be expected for any visually competent system.

**Fine-tuning: Learning is Possible, but Generalization is Severely Limited.** We fine-tuned LLaVA-1.5-7B using LoRA on each of the seven subsets. The resulting cross-evaluation, presented in the heatmap in Figure 3, reveals that the model's ability to generalize is entirely dictated by the underlying visual principles of each test plate. The consistently high accuracy along the diagonal confirms that the model architecture possesses sufficient capacity to master the task when given domain-specific data. However, the off-diagonal performance tells a story of failed generalization.

The analysis reveals a sharp dichotomy in the model's learned abilities. It demonstrates excellent generalization within the "Universal Visibility" group, achieving high accuracy, often above 90%, on cross-task evaluations. For instance, a model trained on 'Viewablebyall' reaches 98.0% on 'BlackWhite'. This is predictable, as all three tasks are solved by detecting luminance contrast, allowing the learned skill to be directly transferred. In stark contrast, generalization among the chrominance-driven categories ('General', 'Protanomaly', 'Protanopia') is poor, with transfer accuracies failing to reach the high levels needed for reliable performance (e.g., as low as 47.5%). This indicates the model is not learning a generalizable rule for red-green perception but is instead overfitting to specific hues. The barrier between these two skill

sets is absolute; transferring a luminance-based skill to a chrominance-based task results in a performance collapse to an accuracy of just 41.2%, confirming that the model learns them as two entirely separate and non-transferable abilities.

Furthermore, the 'Colorblind' or "hidden-digit" category exists as a complete perceptual island. Regardless of the training data, the model's accuracy on this task never rises above near-random chance, further underscoring its literal and inflexible learning patterns. This is consistent with its design, which relies on a unique perceptual artifact that is, by design, inaccessible to the normal trichromatic vision the models have learned to emulate.

**Linear Probing: Poor Generalization Implicates Training Strategies, Not Encoder Limitations.** The brittle generalization observed during fine-tuning led us to question if the bottleneck lies deeper, within the vision encoder itself. We conducted four linear probing experiments, freezing the visual backbone of LLaVA-1.5-7B and Qwen2.5-VL-7B—both with and without their respective projector layers—and training only a single linear layer for classification (Figure 4).

The results from the test set show that, from a practical standpoint, both models exhibit poor generalization ability when only a linear classifier is trained. However, the nature of their failures is diagnostically very different. For LLaVA-1.5, the raw vision encoder's features (without projector) are rich enough to allow the classifier to achieve over 50% test accuracy. While this accuracy is insufficient for a reliable solution, it decisively proves that the general-purpose CLIP encoder successfully captures and preserves the necessary, linearly separable color information. The inclusion of the projector further refines these features, accelerating convergence. Therefore, LLaVA's visual pipeline is fundamentally sound, but a simple linear mapping of its features is not enough to solve the task with high accuracy.

In contrast, the experiments with Qwen2.5-VL reveal a more fundamental problem. Here, the model suffers a catastrophic failure to generalize, with test accuracy remaining near zero in both experiments (with and without the projector). This indicates that the necessary color information is not merely difficult to classify; it appears to be entirely absent from the features produced by the encoder. The key difference is that LLaVA-1.5 uses a general-purpose CLIP encoder, whereas Qwen's was extensively fine-tuned. This stark divergence strongly suggests that Qwen's bespoke visual training, while likely enhancing performance on other benchmarks, has actively damaged its ability to perceive fundamental, low-level color differences. This finding reinforces a critical conclusion: the current paradigm of optimizing for high-level semantic tasks may be creating models that are "intelligent" but perceptually compromised.

### A.3.2 ABLATION STUDIES ON SCALE AND DENSITY

Figure 10 illustrates the performance of three multi-modal large language models—LLaVA-1.5 7B, LLaVA-1.6 7B, and Qwen2.5-VL 7B—under varying conditions of data `density` and `scale`. The analysis reveals distinct and significant trends, particularly for the LLaVA-based architectures.

The left panel of Figure 10 shows a clear *positive correlation* between `density` and `Average Accuracy` for both LLaVA models. As the density increases from 0.3 to 2.0, the performance of LLaVA-1.5 7B and LLaVA-1.6 7B generally improves, peaking at higher density values. This suggests that these models are better able to leverage the provided information when features are more densely concentrated. In stark contrast, the performance of Qwen2.5-VL 7B remains consistently low and is largely unaffected by changes in `density`, indicating a lack of sensitivity to this parameter under our experimental conditions.

Conversely, the right panel reveals a strong *negative correlation* between `scale` and accuracy for the LLaVA models. The highest accuracy is overwhelmingly achieved at the smallest `scale` factor of 0.5, after which performance degrades substantially as the `scale` increases. This indicates a high sensitivity to feature scaling, where larger scales may introduce noise or push feature values into less optimal ranges for

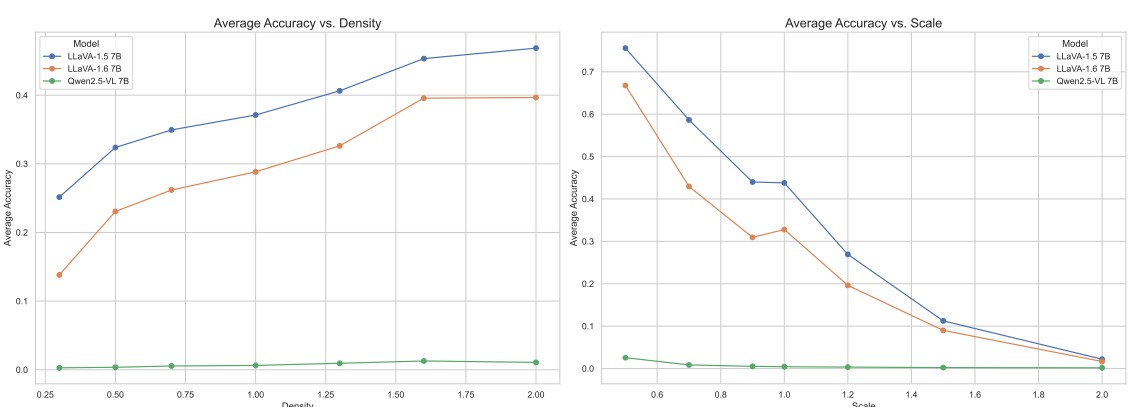

Figure 10: The effect of `density` (left) and `scale` (right) on the average accuracy of different multi-modal large language models. Accuracy is averaged across all tested types ('BlackandWhite', 'Protanopia', and 'Viewablebyall'). The LLaVA models exhibit a strong positive correlation with density and a negative correlation with scale, while the Qwen2.5-VL model shows minimal sensitivity to either parameter.

the models' internal representations. Similar to its behavior with density, the Qwen2.5-VL 7B model shows minimal sensitivity to `scale`, maintaining its low accuracy baseline across the tested range.

In summary, our analysis indicates that the performance of the tested LLaVA models is highly dependent on both data `density` and feature `scale`, strongly favoring high-`density`, low-`scale` configurations. The Qwen2.5-VL model, under these conditions, did not demonstrate comparable performance or sensitivity to these parameters.

## B CONTROLLED COLOR SENSITIVITY TESTS

### B.1 DATA CONSTRUCTION

This document outlines the architecture and workflow of an automated tool designed to generate large-scale datasets of Ishihara-style images. These images, composed of colored dots forming a number (the "figure") against a similarly textured background (the "ground"), are critical for research in color vision deficiency and for training and validating machine learning models designed to recognize patterns under various perceptual conditions.

The tool is built on a procedural generation pipeline that algorithmically controls every aspect of the image creation process, from the spatial distribution of the circles to their precise color properties. This ensures the ability to produce thousands of unique, labeled images with high fidelity and consistency.

The generation of each image follows a distinct, four-step process. This modular approach allows for efficiency and high degrees of customization, as components like the dot pattern can be generated once and reused across multiple color and number configurations. The overall workflow orchestrating these steps is shown in Algorithm 1.

---

**Algorithm 1** High-Level Workflow for Ishihara-style Image Generation

---

**Require:**
    $H, S, V \in [0, 1]$: The base HSV color for the palettes.
    $N \in \mathbb{Z}_{\geq 0}$: The number to display in the image.
**Ensure:**
    $I_{final}$: The final rendered image file.

1: **procedure** GENERATEISHIHARAIMAGE($N, H, S, V$)
2:                                                 ▷ Generate a reusable structural template of circles.
3:     $C_{pattern} \leftarrow$ GENERATESTRUCTURALPATTERN                  ▷ As detailed in Algorithm 2
4:                                                  ▷ Create a binary mask for the figure.
5:     $M_{number} \leftarrow$ CREATEGROUNDTRUTHMASK($N$)           ▷ As detailed in Algorithm 3
6:                                     ▷ Generate distinct color palettes for figure and ground.
7:     $\{\Pi_{on}, \Pi_{off}\} \leftarrow$ GENERATEPARAMETRICPALETTES($H, S, V$)      ▷ As detailed in Algorithm 4
8:                              ▷ Combine pattern, mask, and palettes to render the final image.
9:     $I_{final} \leftarrow$ RENDERFINALIMAGE($C_{pattern}, M_{number}, \Pi_{on}, \Pi_{off}$)      ▷ As detailed in Algorithm 5
10:    Save $I_{final}$ to a file.
11: **end procedure**

---

### B.1.1   STEP 1: STRUCTURAL PATTERN GENERATION

The foundational step in the synthesis of each image is the creation of its structural templateas, as detailed in Algorithm 2. This consists of a dense, non-overlapping pattern of circles designed to fill the entire canvas, meticulously crafted to mimic the distinctive texture of authentic Ishihara plates. To achieve a distribution that is both uniform and devoid of the artificial regularity of a grid, the tool implements a sophisticated method analogous to Poisson Disk Sampling. The algorithm proceeds iteratively, placing circles with randomized radii and positions onto the canvas one by one.

A significant computational challenge in this process is ensuring that no two circles overlap. A naive brute-force approach, which would involve checking a new candidate circle against every circle already placed, is computationally infeasible for patterns comprising tens of thousands of elements. To surmount this performance bottleneck, our algorithm leverages a highly efficient spatial index, specifically a KD-Tree. When considering a new circle for placement, instead of performing a global search, the algorithm queries the KD-Tree to retrieve only the small subset of neighboring circles located in its immediate vicinity. This targeted approach dramatically reduces the number of required intersection tests, transforming the problem's complexity and enabling the rapid generation of dense, intricate patterns.

The final output of this stage is a reusable data structure, denoted as $C_{\text{pattern}}$, which is a comprehensive list containing the coordinates and radius for each circle. This structural template serves as a consistent and reusable foundation, allowing for multiple color schemes and figure configurations to be applied to the same underlying dot pattern, thereby ensuring structural consistency across different test images.

### B.1.2   STEP 2: GROUND TRUTH MASK CREATION

Once the structural pattern of circles is established, the next crucial step is to define the precise shape of the figure to be embedded within it. This is accomplished through the generation of a ground truth template. For any given integer $N$ that is to be displayed in the final image, the tool programmatically constructs a corresponding binary image mask. As detailed in Algorithm 3, this process involves rendering the character or characters for the number $N$ onto the center of a simple, two-dimensional canvas using standard font rendering libraries. The output of this operation is a binary mask, denoted as $M_{\text{number}}$, where all pixels

---

**Algorithm 2** Structural Pattern Generation

---

1: **procedure** GENERATESTRUCTURALPATTERN
2:     Initialize an empty list of circles, $C_{list}$.
3:     Let $N_{total}$ be the target number of circles.
4:     **while** length of $C_{list} < N_{total}$ **do**
5:         Generate a candidate circle, $c_{new}$, with random position and radius.
6:         **if** $C_{list}$ is empty **then**
7:             Add $c_{new}$ to $C_{list}$ and continue.
8:         **end if**
9:         Build a spatial index (e.g., KD-Tree) from the centers of circles in $C_{list}$.
10:         Find a small set of nearest neighbors to $c_{new}$, let this be $C_{neighbors}$.
11:         $has\_overlap \leftarrow$ false.
12:         **for** each circle $c_{neighbor}$ in $C_{neighbors}$ **do**
13:             **if** INTERSECTS($c_{new}, c_{neighbor}$) **then**
14:                 $has\_overlap \leftarrow$ true; **break**.
15:             **end if**
16:         **end for**
17:         **if not** $has\_overlap$ **then**
18:             Add $c_{new}$ to $C_{list}$.
19:         **end if**
20:     **end while**
21:     **return** $C_{list}$.
22: **end procedure**

23: **function** INTERSECTS($c_1, c_2$)
24:     $(x_1, y_1, r_1) \leftarrow c_1$; $(x_2, y_2, r_2) \leftarrow c_2$.
25:     $d^2 \leftarrow (x_2 - x_1)^2 + (y_2 - y_1)^2$.
26:     **return** $d^2 < (r_1 + r_2)^2$.
27: **end function**

---

**Algorithm 3** Ground Truth Mask Creation

---

**Require:**
    $N \in \mathbb{Z}_{\geq 0}$: The number to render.
    $W, H_{canvas}$: Dimensions of the canvas.
**Ensure:**
    $M_{number}$: A binary mask image.
1: **procedure** CREATEGROUNDTRUTHMASK($N$)
2:     Initialize a blank (all zeros) canvas $M_{number}$ of size $W \times H_{canvas}$.
3:     Select a font and font size appropriate for the canvas.
4:     Calculate the position to center the number $N$.
5:     Render the text of number $N$ onto $M_{number}$ with a pixel value of 1.
6:     **return** $M_{number}$.
7: **end procedure**

---

falling within the rendered outline of the number are assigned one value (e.g., 1), while all surrounding background pixels are assigned a contrasting value (e.g., 0). This mask serves as an unambiguous, pixel-perfect map that definitively dictates which circles from the structural pattern belong to the figure and which belong to the background during the final coloring and rendering stage.

**Algorithm 4** Parametric Color Palette Generation

**Require:**
    $H, S, V \in [0, 1]$: The base HSV color.
**Ensure:**
    $\{\Pi_{on}, \Pi_{off}\}$: Two lists of RGB colors.
1:  **procedure** GENERATEPARAMETRICPALETTES($H, S, V$)
2:     Initialize empty color lists $\Pi_{on}$ and $\Pi_{off}$.
3:     **for** a predefined number of 'on' colors **do**
4:         $H' \leftarrow H + \text{random\_jitter}(\sigma_{H,on})$
5:         $S' \leftarrow S + \text{random\_jitter}(\sigma_{S,on})$
6:         $V' \leftarrow V + \text{random\_jitter}(\sigma_{V,on})$
7:         Convert $(H', S', V')$ to RGB and add to $\Pi_{on}$.
8:     **end for**
9:     **for** a predefined number of 'off' colors **do**
10:       $H'' \leftarrow H + \text{random\_jitter}(\sigma_{H,off})$
11:       $S'' \leftarrow S + \text{random\_jitter}(\sigma_{S,off})$
12:       $V'' \leftarrow V + \text{random\_jitter}(\sigma_{V,off})$
13:       Convert $(H'', S'', V'')$ to RGB and add to $\Pi_{off}$.
14:     **end for**
15:     **return** $\{\Pi_{on}, \Pi_{off}\}$.
16: **end procedure**

### B.1.3   Step 3: Parametric Color Palette Generation

This stage represents the core of the tool's powerful color control capabilities. Rather than relying on static, predefined color sets, the palettes for both the figure and the background are generated programmatically from a single base color defined in the HSV (Hue, Saturation, Value) space. The entire procedure is formally detailed in Algorithm 4. Starting with a given base $(H, S, V)$ input, the tool synthesizes two distinct but related sets of colors. The first, designated as the 'ON' palette ($\Pi_{on}$) and intended for the number, is created by applying minor, controlled random variations, or "jitter," to the base color's H, S, and V components. This results in a tight cluster of visually similar colors. Concurrently, a second 'OFF' palette ($\Pi_{off}$) is generated for the background dots by applying a different, and often more significant, set of variations to the same base color. This creates a distinct color cluster that is perceptually related to the first. The final output of this step consists of two lists of colors, $\Pi_{on}$ and $\Pi_{off}$, which have been converted to the RGB color space and are ready for the final rendering process.

### B.1.4   Step 4: Final Image Rendering and Assembly

The final stage of the pipeline is the assembly and rendering process, which synthesizes the outputs from the preceding steps into the complete Ishihara-style image. This procedure, formally described in Algorithm 5, iterates through every single circle defined in the structural template, $C_{\text{pattern}}$. For each circle, a decision is made to classify it as belonging to either the figure or the background. This classification is determined by sampling the corresponding location of the circle's center in the ground truth mask, $M_{\text{number}}$. If the center of a circle falls within the region designated as the number (where the mask's pixel value is 1), that circle is assigned a randomly selected color from the 'ON' palette, $\Pi_{\text{on}}$. Conversely, if the circle's center lies in the background region (where the mask's pixel value is 0), it is assigned a random color from the 'OFF' palette, $\Pi_{\text{off}}$. Each circle is then drawn onto a blank canvas with its assigned color. The culmination of this process is a fully rendered, high-resolution PNG image, where the intended number is subtly yet clearly embedded within the complex field of colored circles.

---

**Algorithm 5** Final Image Rendering and Assembly

---

**Require:**
  $C_{pattern}$: List of circle definitions (position, radius).
  $M_{number}$: Binary mask for the figure.
  $\Pi_{on}, \Pi_{off}$: Color palettes.
**Ensure:**
  $I_{final}$: The final rendered image.
  1: **procedure** RENDERFINALIMAGE($C_{pattern}, M_{number}, \Pi_{on}, \Pi_{off}$)
  2:     Create a blank image, $I_{final}$.
  3:     **for** each circle $c$ in $C_{pattern}$ **do**
  4:         Let $(x_c, y_c)$ be the center of circle $c$.
  5:         **if** the pixel value of $M_{number}$ at $(x_c, y_c)$ is 1 **then**
  6:             $color \leftarrow$ a randomly selected color from $\Pi_{on}$.
  7:         **else**
  8:             $color \leftarrow$ a randomly selected color from $\Pi_{off}$.
  9:         **end if**
  10:        Draw circle $c$ on $I_{final}$ with the selected $color$.
  11:    **end for**
  12:    **return** $I_{final}$.
  13: **end procedure**

---

### B.1.5 UNPARALLELED CUSTOMIZATION VIA PARAMETRIC HSV CONTROL

The key strength of this automation tool lies in its complete and granular control over the color environment of the generated images. By parameterizing the color generation process around a base HSV input, our tool can produce a virtually infinite number of Ishihara-style plates for **any imaginable color scheme**.

The choice of the HSV color space is deliberate and powerful. Unlike RGB, which is machine-oriented, HSV (Hue, Saturation, Value) aligns closely with human perception of color:

- **Hue (H):** The pure color (e.g., red, green, blue).

- **Saturation (S):** The intensity or "purity" of the color, from gray to vibrant.

- **Value (V):** The brightness or darkness of the color.

This perceptual alignment allows us to define the visual distinction between the figure and ground with scientific precision. By controlling the variance applied to each HSV channel independently for the 'ON' and 'OFF' palettes (see Algorithm 4), we can systematically generate datasets that probe specific aspects of vision.

This leads to an unprecedented level of experimental control and dataset diversity:

- **Simulating Specific Deficiencies:** We can precisely simulate common forms of color vision deficiency. Protanopia and deuteranopia (red-green) are modeled by choosing a base hue in the red-to-green range and creating 'ON' and 'OFF' palettes that differ primarily in that hue channel. Similarly, tritanopia (blue-yellow) is modeled using base hues in the blue/yellow spectrum.

- **Creating Perceptual Difficulty Gradients:** The tool can programmatically control the difficulty of each image. By making the statistical distributions of the 'ON' and 'OFF' palettes very close (i.e., small jitter variances), we create a subtle, low-contrast image that is difficult to decipher. By increasing the separation between the palettes, the task becomes easier. This allows for the automated

generation of datasets with a smooth, continuous difficulty parameter, ideal for psychophysics and for training robust machine learning models.

- **Isolating Perceptual Channels:** We can create tests that isolate a single perceptual channel. For instance, a monochromatic test can be generated by setting the hue variation to zero and distinguishing the figure from the ground using only differences in saturation or value. This is impossible to control with such granularity using standard image editing tools.

- **Infinite, Non-Repeating Datasets:** Because the entire process is procedural and randomized (pattern generation, color jitter), the tool can generate a virtually infinite stream of unique, labeled images from a single set of input parameters. This is a critical asset for training deep learning models, as it prevents overfitting on a fixed dataset and provides a limitless source of training, validation, and testing data.

This flexibility and precision make our tool an invaluable asset for building comprehensive, diverse, and perfectly controlled datasets for advanced machine learning and scientific applications.

## B.2 Examples from Controlled Color Sensitivity Tests

We present a series of experiments to analyze variations in color dimensions. In Figure 11, we demonstrate the impact of altering the Hue (H) dimension. For the data presented in the same row, the images differ from each other only in the H dimension. However, there is a clear distinction between the two rows, as the data between them varies across all three HSV (Hue, Saturation, and Value) dimensions.

In Figure 12, we showcase the contrast between an active state ($S_{on}$) and an inactive state ($S_{off}$) at different levels of variance. The first and second rows display reciprocal data, where the roles of $S_{on}$ and $S_{off}$ have been swapped. For human perception, this exchange does not typically increase the difficulty of recognizing the depicted digits. This symmetrical setup, however, can be utilized as a benchmark to evaluate whether a LVLMs possesses a similar sense of symmetry in its analysis.

Furthermore, Figure 13 extends this analysis to the Value dimension. It illustrates the contrast between an active state ($V_{on}$) and an inactive state ($V_{off}$) across various degrees of difference, providing insight into the model's sensitivity to changes in brightness.

## B.3 Detailed Experiment Results and Analysis

### B.3.1 Analysis of Controlled Color Sensitivity Tests in Hue (H) Dimension

An analysis of the performance of several vision-language models across a spectrum of color hues reveals a significant and non-uniform sensitivity to color variations. Figure 5 illustrates the accuracy of models such as LLaVA-1.5-7B, LLaVA-1.5-LoRA-SFT, LLaVA-1.6-7B, Qwen2.5-VL-7B, and Qwen2-VL-7B as a function of the H (Hue) value in the HSV color space. A pronounced trend across most of the evaluated models is a noticeable degradation in performance in the green-yellow portion of the spectrum, specifically around H values of 0.22 and 0.33.

This performance drop suggests that the models' perceptual abilities are not balanced across the entire color gamut. While models like LLaVA-1.5-LoRA-SFT maintain a relatively high accuracy, they are not immune to this dip, indicating a systemic challenge in how these architectures process and interpret certain color ranges. The starkest example of this is the precipitous fall in accuracy for several models when transitioning from blue and red hues to shades of green.

The implications of this color perception imbalance are significant for real-world applications. In scenarios where precise color identification is critical, such as in medical imaging analysis, autonomous vehicle navigation, or content moderation, such a "color blindness" to specific hues could lead to erroneous and

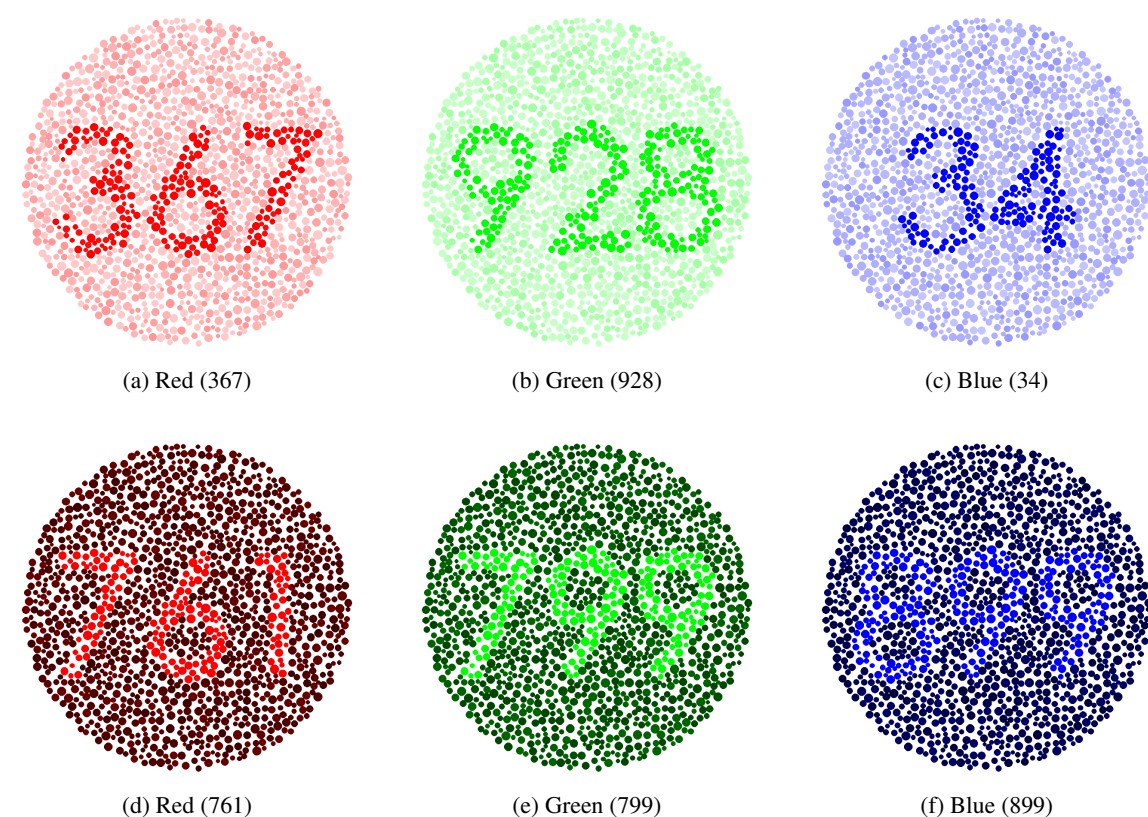

(a) Red (367)  (b) Green (928)  (c) Blue (34)

(d) Red (761)  (e) Green (799)  (f) Blue (899)

Figure 11: Demonstration of Red, Green, and Blue colors with different S and V values.

potentially harmful outcomes. For instance, a self-driving car's inability to consistently recognize a green traffic light under varying lighting conditions could have catastrophic consequences. Similarly, in agricultural applications where crop health is monitored through subtle variations in green, a model's insensitivity could lead to inaccurate assessments.

The newly introduced Qwen2-VL-7B model, while generally exhibiting lower accuracy in this specific test, also conforms to the observed pattern of struggling with the green-yellow spectrum. This consistency across different model architectures and training methodologies points towards a more fundamental issue, possibly rooted in the data distributions of the training sets or inherent biases in the feature extraction processes of these deep learning models. Future research should focus on targeted data augmentation techniques and architectural modifications to mitigate this color perception gap and develop more robust and equitably performing vision-language models. The development of benchmark datasets specifically designed to test for uniform color perception will also be a crucial step in addressing this newly identified challenge.

### B.3.2 ANALYSIS OF CONTROLLED COLOR SENSITIVITY TESTS IN SATURATION (S) DIMENSION

Controlled color sensitivity tests, with results detailed in Table 2, reveal that model accuracy is critically dependent on color contrast, particularly in the Saturation (S) dimension. These experiments, which measure

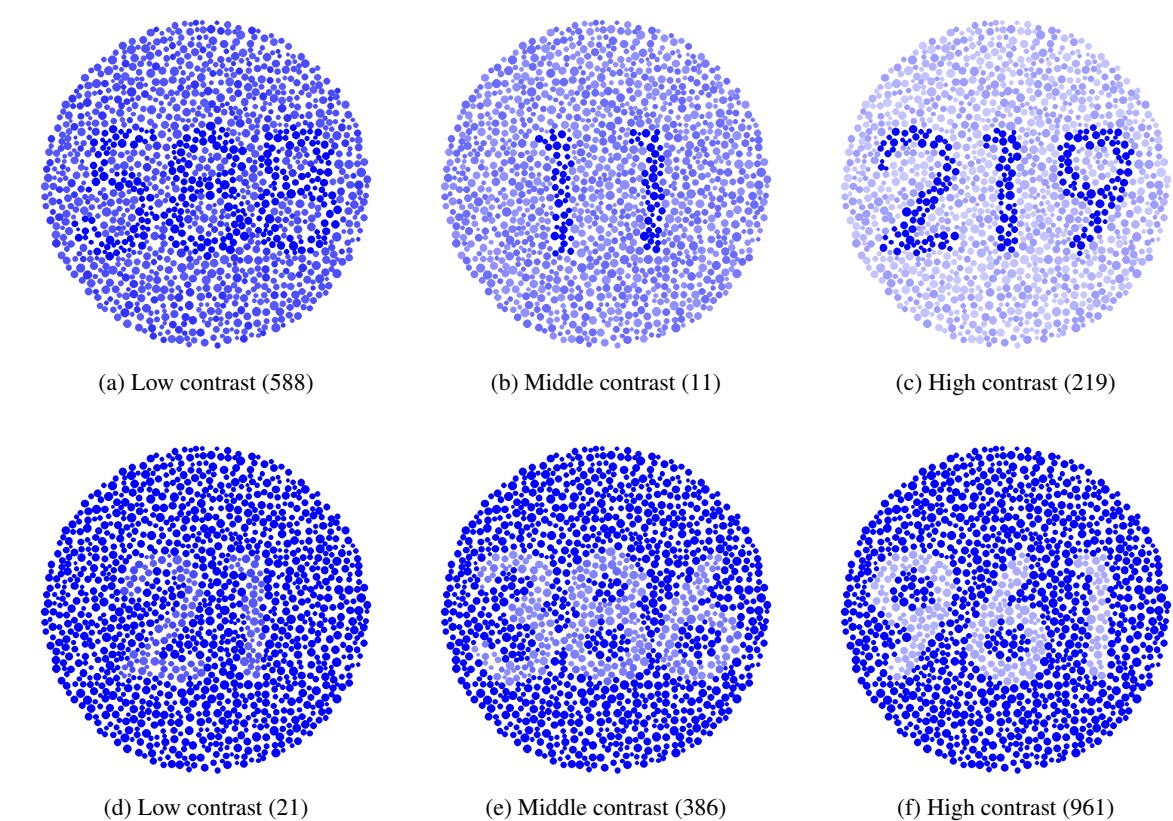

(a) Low contrast (588)  (b) Middle contrast (11)  (c) High contrast (219)

(d) Low contrast (21)  (e) Middle contrast (386)  (f) High contrast (961)

Figure 12: Demonstration of samples with different S values.

performance by systematically altering the saturation of foreground numerals ($S_{On}$) and their backgrounds ($S_{Off}$), uncover significant performance asymmetries not present in human vision.

A fundamental departure from human perception is the models' strong directional bias. For a human observer, the ability to recognize a numeral is largely invariant to a foreground-background color swap; a high-saturation figure on a low-saturation background is typically as legible as the reverse. The models, however, exhibit no such perceptual invariance, showing a profound inability to recognize low-contrast figures against high-contrast backgrounds. This is starkly illustrated by the Qwen2-VL-7B model. It achieves an impressive 86.63% accuracy when identifying a high-saturation numeral on a medium-saturation background ($S_{On} = 1, S_{Off} = 0.3$), yet its performance catastrophically collapses to just 0.43% when the roles are swapped ($S_{On} = 0.3, S_{Off} = 1$).

This extreme reliance on detecting simple "pop-out" signals, rather than discerning figures from a complex chromatic field, is a behavior fundamentally different from the robust figure-ground separation in human vision. A comparative analysis confirms that models are more reliant on high Saturation (S) contrast than on Value (V) contrast. The vast performance gulf for Qwen2-VL-7B between recognizing a high-saturation figure (86.63%) and a low-saturation one (0.43%) highlights that its perceptual system is overwhelmingly tuned to color purity. The failures in the Saturation tests are far more severe across all models, indicating that current LVLMs are more susceptible to variations in color purity than to variations in brightness, exposing a key weakness and a specific area for future improvement in their perceptual training.

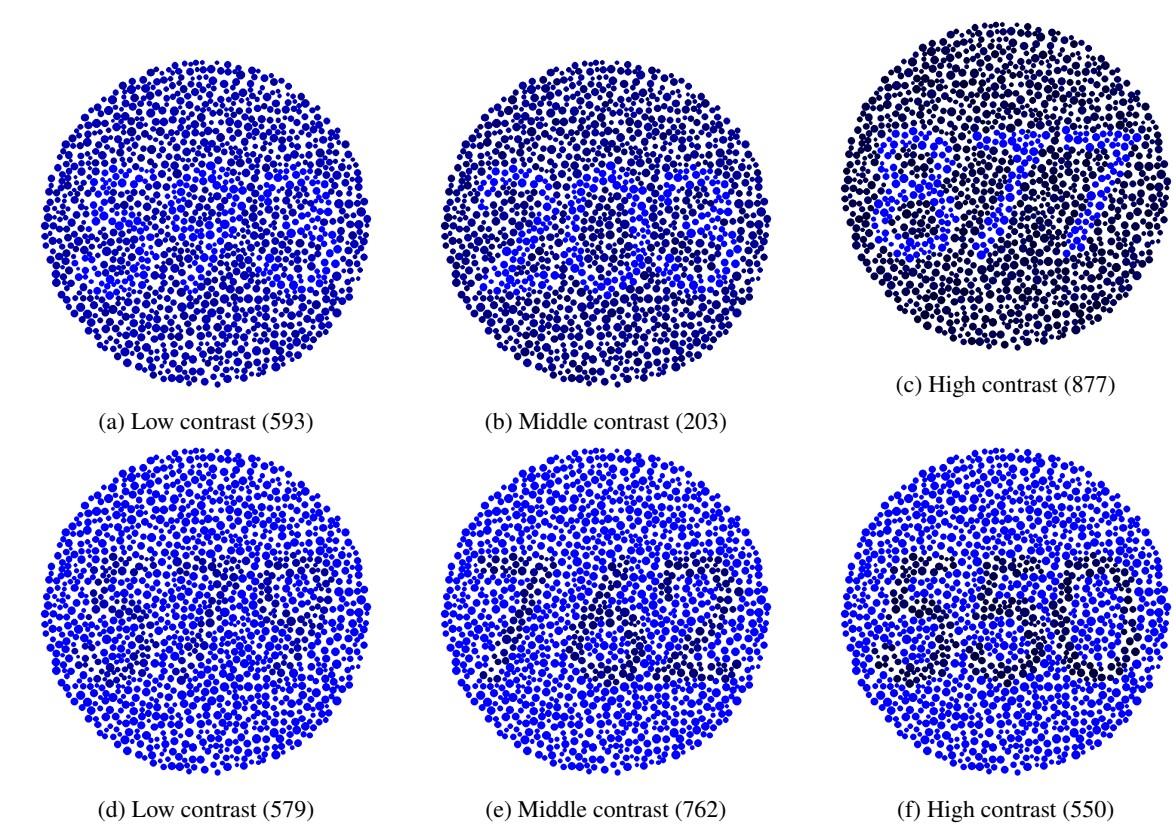

(a) Low contrast (593)   (b) Middle contrast (203)   (c) High contrast (877)

(d) Low contrast (579)   (e) Middle contrast (762)   (f) High contrast (550)

Figure 13: Demonstration of samples with different V values.

### B.3.3 ANALYSIS OF CONTROLLED COLOR SENSITIVITY TESTS IN VALUE (V) DIMENSION

In addition to Hue and Saturation, the Value (V) dimension, or brightness, was analyzed through controlled sensitivity tests, with the results also detailed in Table 2. These experiments, which involved systematically altering the brightness of foreground numerals ("On") and their backgrounds ("Off"), highlight clear robustness gaps across different model families.

The results reveal clear and consistent performance differences between model families in their ability to handle brightness variations. The LLaVA models consistently demonstrate greater robustness than their Qwen counterparts, especially in challenging low-contrast scenarios. For instance, in a difficult low-light condition with a bright numeral on a very dark background ($V_{On} = 1, V_{Off} = 0.1$), the LLaVA 1.5-13B and LLaVA 1.5-7B models maintain respectable accuracies of 48.50% and 49.73% respectively. In stark contrast, the entire Qwen2.5-VL family, including the flagship 32B model, scores near zero under these exact conditions (e.g., 0.17% for Qwen2.5-VL-32B). This significant performance gap suggests that underlying architectural or training differences in the LLaVA family better preserve the ability to process subtle color and brightness variations.

While models exhibit a directional bias in the Value dimension, preferring high-brightness objects on low-brightness backgrounds, this effect is less extreme than the asymmetries observed in the Saturation dimension. Nonetheless, the poor performance of entire model families under specific brightness conditions points

Table 3: Test Results of Different Models (Red/Green/Blue) Under Multi-Parameter Combinations

| S | | V | | Zeroshot | LoRA Models | | |
|---|---|---|---|---|---|---|---|
| On | Off | On | Off | Model | Red | Green | Blue |
| *Red (H=0)* | | | | | | | |
| 1 | 1 | 1 | 0.7 | 0.00 | 95.30* | 0.90 | 2.40 |
| 1 | 1 | 1 | 0.5 | 2.60 | 28.00 | 5.30 | 14.80 |
| 1 | 1 | 1 | 0.3 | 12.40 | 51.30 | 31.90 | 47.50 |
| 1 | 1 | 1 | 0.1 | 25.80 | 58.00 | 45.20 | 58.00 |
| 1 | 0.7 | 1 | 1 | 0.80 | 97.00* | 6.00 | 17.20 |
| 1 | 0.5 | 1 | 1 | 29.70 | 69.10 | 55.10 | 64.10 |
| 1 | 0.3 | 1 | 1 | 60.20 | 73.70 | 68.00 | 70.20 |
| 1 | 0.1 | 1 | 1 | 74.20 | 76.00 | 71.80 | 70.00 |
| *Green (H=0.33)* | | | | | | | |
| 1 | 1 | 1 | 0.7 | 0.00 | 1.00 | 41.90* | 0.90 |
| 1 | 1 | 1 | 0.5 | 1.20 | 10.30 | 7.70 | 11.00 |
| 1 | 1 | 1 | 0.3 | 9.10 | 47.60 | 30.70 | 52.90 |
| 1 | 1 | 1 | 0.1 | 42.30 | 73.30 | 66.70 | 65.50 |
| 1 | 0.7 | 1 | 1 | 0.20 | 8.20 | 46.40* | 4.60 |
| 1 | 0.5 | 1 | 1 | 14.90 | 55.80 | 43.20 | 56.60 |
| 1 | 0.3 | 1 | 1 | 61.10 | 74.40 | 69.60 | 71.50 |
| 1 | 0.1 | 1 | 1 | 75.60 | 77.20 | 74.40 | 74.80 |
| *Blue (H=0.67)* | | | | | | | |
| 1 | 1 | 1 | 0.7 | 0.40 | 6.20 | 1.60 | 98.30* |
| 1 | 1 | 1 | 0.5 | 8.20 | 45.40 | 29.70 | 62.20 |
| 1 | 1 | 1 | 0.3 | 47.40 | 80.40 | 71.60 | 82.70 |
| 1 | 1 | 1 | 0.1 | 61.10 | 79.60 | 78.40 | 82.50 |
| 1 | 0.7 | 1 | 1 | 1.30 | 28.80 | 7.00 | 98.20* |
| 1 | 0.5 | 1 | 1 | 37.60 | 65.80 | 60.50 | 71.60 |
| 1 | 0.3 | 1 | 1 | 67.50 | 76.60 | 71.40 | 75.20 |
| 1 | 0.1 | 1 | 1 | 78.60 | 74.30 | 70.90 | 74.10 |

to a critical area for improvement. This suggests that while models are more sensitive to saturation, significant weaknesses also exist in their ability to perceive and interpret information based on brightness contrast alone, further underscoring the need for more robust perceptual training.

## B.4 INVESTIGATING CROSS-COLOR GENERALIZATION VIA TARGETED LoRA FINE-TUNING

Following our diagnostic findings that highlight models' constrained generalization and biased color perception, we conducted a targeted experiment to assess their ability to transfer learned color discrimination skills across different hue dimensions. This involved fine-tuning LLaVA-1.5-7B using LoRA on specific color-controlled subsets (Red, Green, Blue) and evaluating its performance on other color channels, as detailed in Table 3. The results, also visualized in Figure 14, provide crucial insights into how LVLMs encode and generalize color information.

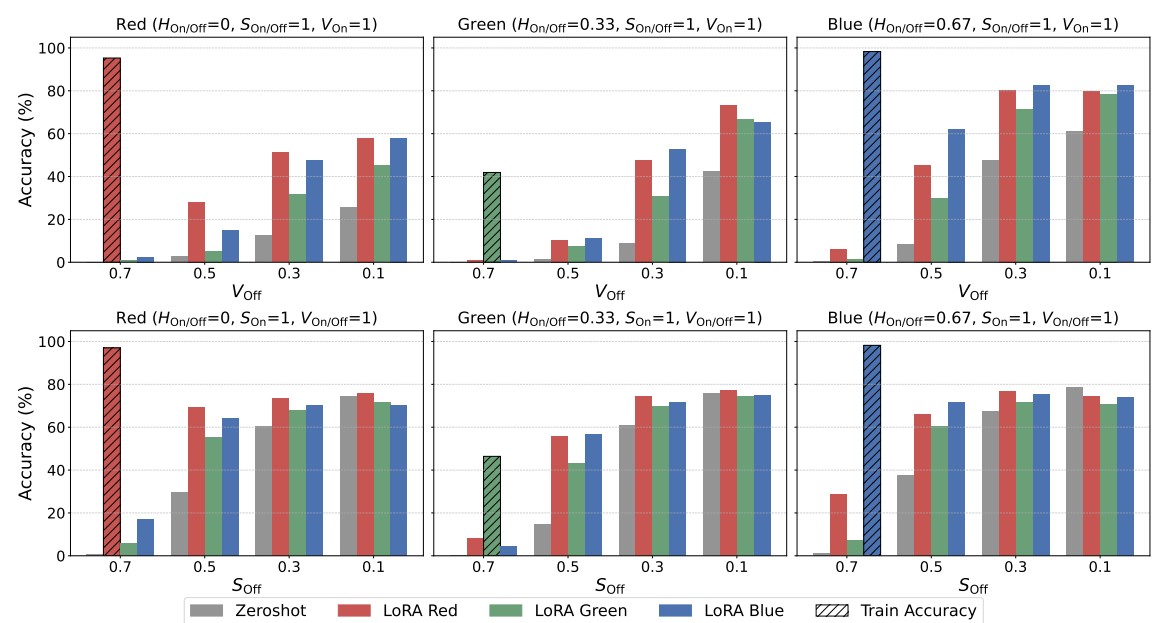

Figure 14: Cross-Color Generalization.

### B.4.1 LoRA Fine-tuning Confirms Color-Specific Learnability but Reveals Weak Cross-Channel Generalization.

The experiment demonstrates that the model is highly capable of learning to discriminate specific color contrasts when explicitly fine-tuned. The accuracies marked with '*' in Table 3 represent the in-domain performance (e.g., model trained on Red data tested on Red data), which consistently shows high scores, often exceeding 95% (e.g., 95.30% for Red, 41.90% for Green, 98.30% for Blue, particularly in higher contrast settings). This confirms that LLaVA-1.5-7B's architecture has the capacity to extract and utilize color-specific features.

However, the primary goal was to assess cross-channel generalization, i.e., whether training on one hue (e.g., Red) enables the model to perceive figures in another (e.g., Green or Blue). The results indicate a pronounced weakness in this area. A model fine-tuned to recognize red figures (e.g., achieving 95.30% on 'Red' with 'V$_{Off}$=0.7' contrast) shows dramatically reduced performance when tested on green (0.90%) or blue (2.40%) figures under similar conditions. This pattern holds across all training hues: training on green significantly boosts performance on green figures but offers minimal benefit for red or blue, and similarly for blue. This stark "diagonal dominance" in the transfer matrix strongly suggests that the model is not learning a universal, hue-agnostic color perception mechanism. Instead, it appears to learn distinct, isolated filters or features for each specific hue, failing to abstract the underlying principle of color contrast in a generalized manner.

### B.4.2 Saturation and Value Sensitivity Further Underline Color-Specific Learning.

The impact of Saturation (S) and Value (V) changes further elucidates this color-specific learning. When contrasting the figures by changing Value ('V$_{On}$=1', 'V$_{Off}$' varies), the accuracy generally increases as the contrast increases (i.e., 'V$_{Off}$' decreases from 0.7 to 0.1). This indicates that the model *can* utilize brightness

differences within each hue channel to improve performance. Similarly, varying Saturation ('$S_{On}$=1', '$S_{Off}$' varies) also shows improved performance with increased saturation contrast. However, crucially, even with strong S or V contrasts, the cross-channel transfer remains poor. For example, a model trained on Red ('H=0') achieves 76.00% on Red when '$S_{On}$=1, $S_{Off}$=0.1', but only 71.80% on Green and 70.00% on Blue under the *same contrast condition*, highlighting that even when luminance or saturation contrast is high, the model's ability to interpret these cues is still heavily gated by the specific hue it was trained on. This reinforces the finding that the model's color perception is fragmented, not unified.

In conclusion, while LoRA fine-tuning effectively enables LLaVA-1.5-7B to learn specific color discrimination tasks, the severe lack of cross-channel generalization points to a fundamental limitation. The model struggles to abstract a general concept of "color contrast" or "figure-ground separation via chrominance" that is independent of the specific hue. Instead, its learning is largely confined to the individual color dimensions it is exposed to during fine-tuning. This fragmentation further explains the model's initial poor performance on the IshiharaColorBench, suggesting that a truly robust color vision system for LVLMs would require training that explicitly fosters a generalized understanding of the entire color space, rather than isolated hue-specific features.

Table 4: HSV Parameter Effects on All 9 Models

| Parameter Group | Qwen Series | | | | | LLaVA Series | | | |
| --- | --- | --- | --- | --- | --- | --- | --- | --- | --- |
| | Qwen2.5-VL | | | Qwen2-VL | | 13B | | 7B | |
| | 32B | 7B | 3B | 7B | 2B | 1.5 | 1.6 | 1.5 | 1.6 |
| **Red (H=0)** | | | | | | | | | |
| S=1, V(On=1, Off=0.7) | 0.00 | 0.00 | 0.00 | 0.10 | 0.10 | 0.50 | 0.10 | 0.40 | 0.30 |
| S=1, V(On=1, Off=0.5) | 0.00 | 0.10 | 0.00 | 0.60 | 1.20 | 12.00 | 6.60 | 8.20 | 3.30 |
| S=1, V(On=1, Off=0.3) | 0.20 | 0.20 | 0.00 | 5.50 | 4.00 | 52.20 | 38.90 | 47.40 | 26.90 |
| S=1, V(On=1, Off=0.1) | 0.00 | 0.30 | 0.10 | 10.80 | 5.60 | 63.70 | 53.30 | 61.10 | 48.30 |
| S=1, V(On=0.7, Off=1) | 0.00 | 0.10 | 0.00 | 0.20 | 0.10 | 0.30 | 0.00 | 0.00 | 0.00 |
| S=1, V(On=0.5, Off=1) | 0.10 | 0.00 | 0.10 | 0.60 | 0.40 | 7.20 | 4.20 | 5.60 | 1.10 |
| S=1, V(On=0.3, Off=1) | 0.10 | 0.10 | 0.00 | 0.90 | 0.40 | 42.10 | 23.90 | 34.70 | 8.50 |
| S=1, V(On=0.1, Off=1) | 0.10 | 0.20 | 0.10 | 2.70 | 0.40 | 55.60 | 38.00 | 53.10 | 25.70 |
| S(On=1, Off=0.7), V=1 | 0.30 | 0.20 | 0.00 | 1.00 | 0.60 | 1.80 | 0.60 | 1.30 | 0.40 |
| S(On=1, Off=0.5), V=1 | 0.30 | 0.30 | 0.00 | 21.60 | 5.80 | 42.80 | 36.40 | 37.60 | 20.50 |
| S(On=1, Off=0.3), V=1 | 5.90 | 1.30 | 0.00 | 92.40 | 76.50 | 70.20 | 67.70 | 67.50 | 69.10 |
| S(On=1, Off=0.1), V=1 | 43.70 | 19.80 | 8.80 | 98.60 | 98.20 | 78.50 | 83.10 | 78.60 | 82.20 |
| S(On=0.7, Off=1), V=1 | 0.00 | 0.10 | 0.00 | 0.00 | 0.20 | 0.40 | 0.10 | 0.10 | 0.00 |
| S(On=0.5, Off=1), V=1 | 0.20 | 0.10 | 0.10 | 0.20 | 0.30 | 9.50 | 5.70 | 8.30 | 3.90 |
| S(On=0.3, Off=1), V=1 | 0.10 | 0.00 | 0.00 | 0.40 | 0.10 | 49.20 | 34.60 | 41.60 | 29.30 |
| S(On=0.1, Off=1), V=1 | 0.20 | 0.20 | 0.10 | 4.10 | 0.50 | 61.70 | 50.10 | 59.20 | 53.30 |
| **Green (H=0.33)** | | | | | | | | | |
| S=1, V(On=1, Off=0.7) | 0.00 | 0.20 | 0.00 | 0.20 | 0.10 | 0.10 | 0.10 | 0.00 | 0.10 |
| S=1, V(On=1, Off=0.5) | 0.10 | 0.00 | 0.00 | 0.60 | 0.40 | 1.70 | 0.80 | 1.20 | 0.70 |
| S=1, V(On=1, Off=0.3) | 0.20 | 0.10 | 0.30 | 1.30 | 0.90 | 13.50 | 5.80 | 9.10 | 5.60 |
| S=1, V(On=1, Off=0.1) | 0.20 | 0.10 | 0.10 | 1.70 | 1.20 | 51.70 | 34.40 | 42.30 | 29.80 |
| S=1, V(On=0.7, Off=1) | 0.10 | 0.20 | 0.10 | 0.20 | 0.20 | 0.20 | 0.10 | 0.10 | 0.10 |
| S=1, V(On=0.5, Off=1) | 0.10 | 0.10 | 0.10 | 0.50 | 0.20 | 4.00 | 2.20 | 2.70 | 1.00 |
| S=1, V(On=0.3, Off=1) | 0.00 | 0.10 | 0.00 | 4.00 | 0.80 | 23.30 | 17.10 | 20.10 | 6.70 |
| S=1, V(On=0.1, Off=1) | 0.10 | 0.30 | 0.00 | 26.70 | 4.20 | 48.30 | 39.80 | 44.80 | 24.70 |
| S(On=1, Off=0.7), V=1 | 0.20 | 0.30 | 0.00 | 0.60 | 0.20 | 0.40 | 0.20 | 0.20 | 0.20 |
| S(On=1, Off=0.5), V=1 | 0.30 | 0.10 | 0.20 | 19.20 | 1.20 | 22.00 | 13.60 | 14.90 | 8.90 |
| S(On=1, Off=0.3), V=1 | 2.90 | 1.80 | 1.10 | 86.90 | 51.30 | 64.00 | 57.10 | 61.10 | 59.60 |
| S(On=1, Off=0.1), V=1 | 44.60 | 24.30 | 15.50 | 98.10 | 97.60 | 77.40 | 79.30 | 75.60 | 78.80 |
| S(On=0.7, Off=1), V=1 | 0.00 | 0.10 | 0.00 | 0.20 | 0.10 | 0.10 | 0.10 | 0.30 | 0.10 |
| S(On=0.5, Off=1), V=1 | 0.10 | 0.00 | 0.10 | 0.20 | 0.20 | 2.40 | 1.20 | 2.50 | 1.30 |
| S(On=0.3, Off=1), V=1 | 0.20 | 0.00 | 0.00 | 0.70 | 0.30 | 28.80 | 24.40 | 23.70 | 21.30 |
| S(On=0.1, Off=1), V=1 | 0.10 | 0.30 | 0.10 | 5.00 | 0.20 | 54.60 | 45.40 | 51.70 | 53.60 |
| **Blue (H=0.67)** | | | | | | | | | |
| S=1, V(On=1, Off=0.7) | 0.10 | 0.10 | 0.10 | 0.00 | 0.00 | 0.00 | 0.20 | 0.00 | 0.20 |
| S=1, V(On=1, Off=0.5) | 0.10 | 0.20 | 0.10 | 0.40 | 0.30 | 2.20 | 1.20 | 2.60 | 0.90 |
| S=1, V(On=1, Off=0.3) | 0.10 | 0.00 | 0.20 | 1.70 | 0.70 | 15.80 | 10.00 | 12.40 | 6.60 |
| S=1, V(On=1, Off=0.1) | 0.50 | 0.10 | 0.20 | 3.40 | 1.60 | 30.10 | 22.30 | 25.80 | 14.40 |
| S=1, V(On=0.7, Off=1) | 0.00 | 0.10 | 0.10 | 0.00 | 0.40 | 0.10 | 0.00 | 0.10 | 0.10 |
| S=1, V(On=0.5, Off=1) | 0.20 | 0.00 | 0.10 | 0.30 | 0.30 | 2.90 | 1.20 | 2.20 | 0.70 |
| S=1, V(On=0.3, Off=1) | 0.20 | 0.20 | 0.20 | 0.50 | 0.30 | 16.00 | 9.20 | 11.90 | 4.40 |
| S=1, V(On=0.1, Off=1) | 0.30 | 0.30 | 0.10 | 1.50 | 0.30 | 36.70 | 24.60 | 28.20 | 11.50 |
| S(On=1, Off=0.7), V=1 | 0.00 | 0.00 | 0.00 | 0.50 | 0.20 | 0.70 | 0.20 | 0.80 | 0.20 |
| S(On=1, Off=0.5), V=1 | 0.30 | 0.20 | 0.10 | 15.10 | 6.10 | 35.70 | 19.80 | 29.70 | 16.40 |
| S(On=1, Off=0.3), V=1 | 8.00 | 2.70 | 1.10 | 80.60 | 79.90 | 70.70 | 59.00 | 60.20 | 66.10 |
| S(On=1, Off=0.1), V=1 | 49.40 | 26.20 | 12.70 | 98.30 | 97.70 | 78.20 | 78.20 | 74.20 | 77.80 |
| S(On=0.7, Off=1), V=1 | 0.10 | 0.10 | 0.00 | 0.00 | 0.00 | 0.20 | 0.10 | 0.20 | 0.20 |
| S(On=0.5, Off=1), V=1 | 0.20 | 0.30 | 0.30 | 0.00 | 0.10 | 6.90 | 3.60 | 6.60 | 3.50 |
| S(On=0.3, Off=1), V=1 | 0.00 | 0.10 | 0.00 | 0.20 | 0.00 | 41.10 | 24.60 | 36.10 | 23.30 |
| S(On=0.1, Off=1), V=1 | 0.10 | 0.00 | 0.10 | 3.00 | 0.60 | 52.60 | 37.10 | 55.70 | 42.10 |

