# OpenReview forum: "Color Blindness Test Images as Seen by Large Vision-Language Models"
_ICLR.cc/2026/Conference — Submitted to ICLR 2026_

### Official Review · Reviewer_DH2F · 2025-10-18

**Soundness:** 2
**Presentation:** 2
**Contribution:** 1
**Rating:** 0
**Confidence:** 4

**Summary:**

The paper introduces IshiharaColorBench, a benchmark designed to evaluate Large Vision-Language Models (LVLMs) using traditional Ishihara plates. The benchmark consists of around 7,000 images representing digits from 0 to 999, with colors systematically altered along multiple dimensions. Using this dataset, the authors conduct a comprehensive evaluation of various VLMs. The results show that LVLMs exhibit significantly weaker color perception compared to humans, and even fine-tuning fails to generalize well. This is attributed to models relying on semantic associations rather than genuine color understanding. Furthermore, the Controlled Color Sensitivity Tests reveal clear non-human-like biases, including a particular weakness in perceiving green tones and an over-reliance on saturation contrast.

**Strengths:**

- The paper conducts a comprehensive evaluation across a wide range of LVLMs

**Weaknesses:**

- The colour-blind test setting is already included as a subset of ColorBench (Liang et al.), so the novelty of this benchmark is somewhat limited.
- It is unclear whether the proposed setup truly isolates color perception from semantic understanding. To disentangle semantic and perceptual factors, one might expect the model to first correctly identify digits in the Number Only case with digits shown in black (which might be the normal case in general), and only then be tested under color variations. Without such control, it is still difficult to claim that the results purely reflect color sensitivity rather than semantic cues.
- The paper does not clearly specify how the questions are formatted. For instance, whether the task is multiple-choice or open-ended, and how exactly the input prompt is constructed for LVLMs.
- Some of the analytical findings, such as the weakness in perceiving green tones and the non-continuous color representation space, have already been discussed in prior work, notably in [A] Hyeon-Woo et al., “VLM’s Eye Examination: Instruct and Inspect Visual Competency of Vision-Language Models.” Hence, the paper’s analytical novelty is somewhat limited.

**Questions:**

- The paper shows that high performance on general tasks does not necessarily translate to strong performance on colour-blind tests. In contrast, would improving performance on IshiharaColorBench in turn enhance general visual understanding or lead to tangible downstream benefits? The practical motivation and implications of optimizing for this benchmark are therefore somewhat ambiguous.
- The results indicate that simple fine-tuning does not generalize well to unseen color variations. This raises the question of how color-blind performance could be improved. The paper would have been stronger with some discussion on potential future directions.

---

> ### Author Response · Authors · 2025-12-03
>
> >The colour-blind test setting is already included as a subset of ColorBench (Liang et al.), so the novelty of this benchmark is somewhat limited.
>
> Thank you for your suggestions. While we address this comparison in our related work section, we want to re-emphasize the crucial distinctions, as the assertion of limited novelty misunderstands the different scientific goals of the two projects. ColorBench is a valuable, comprehensive benchmark designed to assess a broad range of color skills, primarily within the context of natural images where color and semantics are inherently entangled. In stark contrast, our work is a targeted, diagnostic benchmark motivated by a completely different objective: to rigorously isolate color perception from semantic priors. The inclusion of a superficially similar subset in a broader benchmark does not diminish the novelty of a project entirely dedicated to this principle of semantic isolation, which allows for a far deeper and more systematic analysis of the models' foundational perceptual failures and biases. Ultimately, the two are not competitors but complementary works asking different scientific questions. ColorBench asks, "How well do models handle various color tasks?", whereas we ask the more fundamental question, "Do models genuinely perceive color, or are they just exploiting learned semantic shortcuts?"
>
>
> >It is unclear whether the proposed setup truly isolates color perception from semantic understanding. To disentangle semantic and perceptual factors, one might expect the model to first correctly identify digits in the Number Only case with digits shown in black (which might be the normal case in general), and only then be tested under color variations. Without such control, it is still difficult to claim that the results purely reflect color sensitivity rather than semantic cues.
>
> Thank you for your suggestions. This concern is addressed by a control experiment that was indeed performed, which we will clarify in the revised manuscript. We tested all models on a baseline task of recognizing standard black digits on a white background, and every model achieved 100% accuracy. This result unequivocally establishes that the models possess the basic OCR capabilities and can follow the instructions for the task.
>
>
> Having established this baseline, our NumberOnly control (using monochromatic dots) was designed to isolate the next variable: the difficulty of perceiving a digit from a procedural dot pattern. While performance drops from the perfect baseline, the models are still largely successful, proving they can handle the pattern recognition aspect of the task.
>
>
> Therefore, the catastrophic failure observed on the color plates can only be attributed to the final variable we introduce: color difference. Our multi-stage experimental design—from simple OCR to monochromatic patterns to chromatic patterns—is specifically structured to systematically eliminate confounding factors. This allows us to conclude with high confidence that the dramatic performance drop is rooted in a fundamental limitation in color perception, not in an inability to understand the task or process the visual texture.
>
>
> >The paper does not clearly specify how the questions are formatted. For instance, whether the task is multiple-choice or open-ended, and how exactly the input prompt is constructed for LVLMs.
>
> As shown in Figure 1, the evaluation is conducted using an open-ended visual question answering format, not multiple-choice.

---

> > ### Author Response · Authors · 2025-12-03
> >
> > >Some of the analytical findings, such as the weakness in perceiving green tones and the non-continuous color representation space, have already been discussed in prior work, notably in [A] Hyeon-Woo et al., “VLM’s Eye Examination: Instruct and Inspect Visual Competency of Vision-Language Models.” Hence, the paper’s analytical novelty is somewhat limited.
> >
> > We would like to directly address the comment on analytical novelty. While both our work and [A] observe a weakness in perceiving green tones, we believe this single, high-level observation is where the similarity ends. Our paper's novelty lies not in simply re-identifying a color bias, but in uncovering the underlying mechanisms and functional consequences of this failure with a new level of detail. Specifically, our analysis is the first to demonstrate the models' systemic inability to distinguish information based on brightness contrast while retaining some capability with saturation contrast—a more fundamental finding about how their perception operates. Furthermore, we move beyond noting a "sensitivity" by proving that this weakness leads to a catastrophic task failure (performance close to random guessing) in a controlled, semantically-neutral environment. Finally, we establish that this is a deep-seated limitation by showing that neither scaling-up nor fine-tuning leads to generalizable improvement. These deeper insights, enabled by our specific methodology, represent a distinct and complementary contribution to the understanding of VLM perception.
> >
> >
> > >The paper shows that high performance on general tasks does not necessarily translate to strong performance on colour-blind tests. In contrast, would improving performance on IshiharaColorBench in turn enhance general visual understanding or lead to tangible downstream benefits? The practical motivation and implications of optimizing for this benchmark are therefore somewhat ambiguous.
> >
> > Thank you for your suggestions. The primary motivation for IshiharaColorBench is not to serve as a fine-tuning dataset that promises direct, incremental gains on existing downstream tasks. Rather, our objective is to diagnose a critical and overlooked deficiency in current state-of-the-art models. Our findings demonstrate that popular leaderboards, by not explicitly testing for this kind of foundational perception, inadvertently allow models to succeed by exploiting semantic shortcuts. The tangible implication of our work is therefore to provide a clear direction for improving future evaluations. By revealing this systemic blind spot, we provide the community with the tools and the motivation to build more comprehensive benchmarks. The ultimate goal is to encourage the development of models that can genuinely perceive color, leading to more robust and reliable visual systems, which is a necessary step before true downstream benefits can be consistently realized.
> >
> >
> > >The results indicate that simple fine-tuning does not generalize well to unseen color variations. This raises the question of how color-blind performance could be improved. The paper would have been stronger with some discussion on potential future directions.
> >
> > Thank you for your suggestions. The primary contribution of this paper is to provide a rigorous diagnosis of this fundamental limitation. We agree that exploring effective remedies is the crucial next step. We are actively planning to investigate potential solutions to improve this phenomenon in our follow-up research and will add a note to the conclusion of our revised manuscript to reflect this important direction for future work.

---

### Official Review · Reviewer_esMf · 2025-10-31

**Soundness:** 2
**Presentation:** 2
**Contribution:** 2
**Rating:** 2
**Confidence:** 3

**Summary:**

This paper investigates whether large vision-language models (LVLMs) genuinely perceive colors like humans or merely rely on semantic correlations. The authors introduce IshiharaColorBench, inspired by human color blindness tests, to directly assess LVLMs’ true color perception. They design two evaluations: standard color blindness tests for performance measurement and controlled color sensitivity tests for behavioral analysis. Results show that LVLMs perform near random guessing and exhibit systematic biases in hue and saturation perception, revealing major limitations in their genuine color understanding and highlighting the need for more perceptually grounded model designs.

**Strengths:**

- The authors introduce IshiharaColorBench, inspired by human color blindness tests, to directly assess LVLMs’ true color perception

- Based on two settings, the authors investigate the (in)ability of LVLMs' color perception capability

**Weaknesses:**

- I completely agree that evaluating the true visual perception capability of LVLMs is essential. However, in my opinion, this paper lacks novelty and interest, as there are already two existing works [1,2] that explore similar perception settings. Therefore, even if the idea itself is not novel, the authors should clearly describe how their benchmark differs from these prior ones. Although Lines 071–079 do provide some comparisons with existing benchmarks, the distinction should be described more explicitly in comparison with [1,2].

- I highly appreciate the large number of experiments conducted — it is indeed impressive. However, since the results appear quite random, I understand that identifying consistent patterns and conducting in-depth analysis must have been challenging for the authors. Nevertheless, rather than stopping at a superficial interpretation such as “there is no scaling law” or “latest models perform poorly,” the paper should include a deeper analysis or discussion on why such phenomena occur. For example, one possible explanation could be that the latest models, during post-training, have overemphasized reasoning datasets, which might have led to catastrophic forgetting of perception capabilities.

- The paper’s readability could be improved. For instance, essential details such as dataset construction and evaluation metrics — even in brief form — should be included in the main text. Currently, since the dataset construction process is only found in the appendix, it was difficult to follow. Additionally, the font size in Figure 5 is too small.

---

References:

[1] HueManity: Probing Fine-Grained Visual Perception in MLLMs

[2] ColorBlindnessEval: Can Vision-Language Models Pass Color Blindness Tests?

**Questions:**

See Weaknesses.

---

> ### Author Response · Authors · 2025-12-03
>
> >I completely agree that evaluating the true visual perception capability of LVLMs is essential. However, in my opinion, this paper lacks novelty and interest, as there are already two existing works [1,2] that explore similar perception settings. Therefore, even if the idea itself is not novel, the authors should clearly describe how their benchmark differs from these prior ones. Although Lines 071–079 do provide some comparisons with existing benchmarks, the distinction should be described more explicitly in comparison with [1,2].
>
> Thank you for your suggestions. Our work is fundamentally about independently evaluating color perception in LVLMs, i.e., determining whether models can truly process chromatic information without leaning on correlations with other semantics. While we, [1], and [2] all use color-blindness plates, we do so for different purposes. HueManity [1] leverages such plates mainly to stress-test OCR robustness under chromatic camouflage, and ColorBlindnessEval [2] provides a preliminary check on whether models “seem color-blind,” without explicitly disentangling color from other cues. In our study, the plates are treated purely as instruments to isolate color, and IshiharaColorBench is constructed so that a digit is recognizable if and only if the model genuinely perceives color. We complement standard tests with controlled sensitivity analyses (e.g., separating saturation from brightness contrasts) to expose systematic biases such as red preference and insensitivity to brightness contrast. We will add a dedicated, side-by-side comparison with [1,2] in the revised manuscript to make these distinctions explicit in terms of motivation, design, and evaluation protocol. Importantly, we do not claim the dataset itself as our novelty; it is a tool. The contribution lies in showing whether color can be evaluated as an independent semantic channel, and in revealing how current LVLMs understand color and where they fall short relative to human perception.
>
> >I highly appreciate the large number of experiments conducted — it is indeed impressive. However, since the results appear quite random, I understand that identifying consistent patterns and conducting in-depth analysis must have been challenging for the authors. Nevertheless, rather than stopping at a superficial interpretation such as “there is no scaling law” or “latest models perform poorly,” the paper should include a deeper analysis or discussion on why such phenomena occur. For example, one possible explanation could be that the latest models, during post-training, have overemphasized reasoning datasets, which might have led to catastrophic forgetting of perception capabilities.
>
> Thank you for your suggestions and we will expand both the empirical and explanatory components. Concretely, we plan to probe whether post-training on reasoning-heavy corpora induces catastrophic forgetting of low-level chromatic cues by correlating post-training mixtures with color performance and by conducting controlled fine-tuning experiments that reintroduce color-focused supervision. We will examine representation bottlenecks by comparing encoders and adapters that differ in how they normalize or compress color channels, and by testing hue-rotation, grayscale, and saturation/brightness-controlled variants to separate reliance on semantic priors from genuine chromatic processing. We will also analyze systematic color biases and their interaction with learned priors, and include qualitative evidence such as saliency maps to illustrate failure modes when color is the only informative signal. These additions aim to move from surface-level findings to principled explanations grounded in training dynamics and architectural choices.
>
> >The paper’s readability could be improved. For instance, essential details such as dataset construction and evaluation metrics — even in brief form — should be included in the main text. Currently, since the dataset construction process is only found in the appendix, it was difficult to follow. Additionally, the font size in Figure 5 is too small.
>
> On readability, we will rebalance the presentation so that essential dataset and metric details are accessible in the main text while full specifics remain in the appendix. We will add a concise description of the construction pipeline, plate variants, and the evaluation metrics we use (accuracy, confusion patterns, and sensitivity curves for saturation and brightness contrasts). We will improve figure readability by enlarging the font size in Figure 5, refining captions, and tightening cross-references so readers can follow the setup without leaving the main narrative. Although our emphasis is on the findings and interpretations rather than the dataset contribution, we agree that presenting these essentials up front will make the paper easier to follow and more informative.

---

### Official Review · Reviewer_wJU5 · 2025-10-31

**Soundness:** 2
**Presentation:** 3
**Contribution:** 1
**Rating:** 2
**Confidence:** 4

**Summary:**

This paper investigates whether Large Vision-Language Models (LVLMs) genuinely perceive color in a human-like way or merely rely on semantic associations between color and object identity.
The authors introduce IshiharaColorBench, a benchmark inspired by the medical Ishihara test, explicitly designed to disentangle true color perception from semantic priors. It includes two types of evaluations: 1) Standard Color Blindness Tests to assess performance, and 2) Controlled Color Sensitivity Tests to analyze perceptual biases along hue (H), saturation (S), and value (V) dimensions.

Experimental results across state-of-the-art LVLMs (GPT-4o, Gemini 2.0 Flash, LLaVA, Qwen2.5-VL, InternVL) reveal that even the best models perform near chance levels on color-based recognition tasks that humans solve perfectly. Scaling and fine-tuning yield little generalization.
Controlled analyses uncover systematic non-humanlike biases.
The paper concludes that current LVLMs lack genuine, low-level color perception and instead rely on semantic shortcuts.

**Strengths:**

- Proposes the benchmark explicitly isolating chromatic perception from semantics in LVLMs

- Respective analyses are interesting and may be helpful to report in the community.

- Comprehensive experimental design, testing across 7,000 procedurally generated images and multiple LVLM architectures.

- Diverse experiments, including LoRA fine-tuning, linear probing, and HSV-controlled sensitivity analyses, which provide quantitative and qualitative insights into perceptual failure modes.

- Reproducibility seems to be well supported: full procedural generation algorithms and evaluation details are provided (Algorithms 1–5, Appendices A–B).

**Weaknesses:**

- Missing important reference: The authors missed the critical citations [C1-3] that are closely related to the submission. In particular, [C1] proposed a very similar benchmark based on Ishihara test.

- Missing statistical significance tests: formal statistical tests or confidence intervals are absent, which would make the findings and conclusions more rigorous.

- Potential confounding factors in image complexity: Though IshiharaColorBench isolates color semantics, the procedural textures might introduce visual noise affecting model OCR components; disentangling this effect could clarify whether failures stem purely from color perception or from pattern segmentation challenges.

- Overly aligning the human test and the machine vision test: The way of human percieve the color and that of the machine are obviously different. This raises the question, why the Ishihara test should be particularly used for the purpose in this work.

- Line 045 "common-sense expection": This review disagrees about this statement. It is unrealistic to expect VLMs to behave exactly like humans as a common-sense baseline. The paper should clarify what level of human-like performance is reasonable to expect and why.

- Line 176 states "the Ishihara plates as a scientifically validated paradigm," but this test has only been validated for color blindness detection. It cannot be generalized to assess VLM color accuracy as targeted in this paper.

- Weak motivation: The overall motivation needs to be strengthened, particularly regarding why color naming is a critical problem. While the authors mention several important cases (Lines 101-102), these cover only a very narrow scope. In fact, most real-world visual perception tasks can be adequately solved with prototypical color perception abilities limited to a few basic categories such as "reddish," "bluish," "greenish," "white," and "black," except for the few counterexamples the authors cited.

- The paper's motivation, scope, and practical impact regarding why color naming is important remain weak. The work would have been better aligned with its analyses if it had presented clear applications, such as computational perceptual quality assessment in color and colorimetry domains, similar to recent "VLM as judge" approaches.

- Restricted exploration of detail causes and remedies: The paper diagnoses the failure convincingly but offers limited discussion or experiments on first identifying causes (where does the issue come from? visual encoder, LLM decoder, training datset, or training procedure?) and the possible architectural or training modifications that could mitigate the issue.

- This reviewer expected the paper to identify which components of VLMs constitute the bottleneck for color naming performance, but such analysis is absent. While the current analyses are well-executed, the paper would be significantly strengthened by including investigations that pinpoint specific bottlenecks in the VLM architecture or processing pipeline.

- Line 107 appears to be an overclaim. Color bias issues and hallucinations have already been documented in prior work (e.g., [C3-5]), so these are not entirely new findings or illusions that were previously unknown.

**Questions:**

- Lines 071-079 discuss VLMs' genuine understanding of color, but the proposed benchmark and analysis methods do not align well with this goal. Specifically, is the Ishihara test truly capable of assessing true color understanding? The generated data contains luminance variations, preventing consistent representation of absolute colors. Moreover, the original Ishihara test evaluates not only color hue but also the ability to perceive relative color differences. Therefore, the Ishihara test alone cannot adequately assess the absolute color awareness that the authors intend to measure. Additional absolute color measurement assessment methods should be employed in parallel to fully support the paper's argument.


- Section 4.2: Green insensitivity has also been mentioned in [C3]. Although the experimental approach differs, what is the distinction between the findings in that paper and those presented in this section?

---

> ### Author Response · Authors · 2025-12-03
>
> >Missing important reference: The authors missed the critical citations [C1-3] that are closely related to the submission. In particular, [C1] proposed a very similar benchmark based on Ishihara test.
>
> >Section 4.2: Green insensitivity has also been mentioned in [C3]. Although the experimental approach differs, what is the distinction between the findings in that paper and those presented in this section?
>
> >Line 107 appears to be an overclaim. Color bias issues and hallucinations have already been documented in prior work (e.g., [C3-5]), so these are not entirely new findings or illusions that were previously unknown.
>
> Thank you for your suggestions. Regarding the questions related to references [C1-C5], the full citations were not provided. We would be grateful if you could share them so we can address your points.
>
> >Missing statistical significance tests: formal statistical tests or confidence intervals are absent, which would make the findings and conclusions more rigorous.
>
> Thank you for your suggestions. We will add statistical significance tests and confidence intervals to our results in the revised manuscript to improve the rigor of our conclusions.
>
>
> >Potential confounding factors in image complexity: Though IshiharaColorBench isolates color semantics, the procedural textures might introduce visual noise affecting model OCR components; disentangling this effect could clarify whether failures stem purely from color perception or from pattern segmentation challenges.
>
> Thank you for your suggestions. This is a valid consideration, which is precisely why we designed our fine-tuning experiments to disentangle these factors. Our results demonstrate two key findings: first, that models can successfully learn the procedural pattern when fine-tuned, as shown by the improved in-distribution performance. This confirms that the pattern itself is not an insurmountable segmentation challenge. Second, and more critically, this learned ability fails to generalize to images with novel color combinations. This specific failure mode—where the pattern is known but the colors are new—allows us to conclude that the bottleneck is the model's limited color perception, not its ability to handle the visual texture.
>
>
> >Overly aligning the human test and the machine vision test: The way of human percieve the color and that of the machine are obviously different. This raises the question, why the Ishihara test should be particularly used for the purpose in this work.
>
> Thank you for your suggestions. We would like to clarify our methodological choice. Our work is built on the established premise that human and machine perception are fundamentally different. For this reason, we selected the Ishihara test not to anthropomorphize models, but to employ it as a precise diagnostic tool. Its unique design is critical because it isolates color perception from the semantic and contextual shortcuts prevalent in natural images, creating a controlled environment where a model's underlying perceptual strategies are unavoidably exposed. The test's purpose in our work is therefore not to see if machines are "human-like," but to rigorously uncover the non-human and brittle nature of their color processing capabilities, which is a goal that standard benchmarks cannot achieve.
>
> >Line 045 "common-sense expection": This review disagrees about this statement. It is unrealistic to expect VLMs to behave exactly like humans as a common-sense baseline. The paper should clarify what level of human-like performance is reasonable to expect and why.
>
> Thank you for your suggestions. We will clarify our position. The expectation is not that LVLMs should mimic human perception, but that they should demonstrate functional competence on a fundamental visual task. Therefore, the reasonable performance baseline we require is one that is significantly and reliably above random chance.
>
> This standard is justified because any model claiming general visual understanding must not fail catastrophically on basic perception when its learned semantic shortcuts are removed. Performing at chance-level, as we show, indicates a foundational failure in perception, suggesting the model's success elsewhere is not based on genuine sight but on exploiting learned correlations. Our test is designed precisely to expose this critical gap.

---

> > ### Author Response · Authors · 2025-12-03
> >
> > >Line 176 states "the Ishihara plates as a scientifically validated paradigm," but this test has only been validated for color blindness detection. It cannot be generalized to assess VLM color accuracy as targeted in this paper.
> >
> > Thank you for your suggestions. This comment mischaracterizes our evaluation metric. Our paper does not directly measure "color accuracy." Instead, we measure the accuracy of digit recognition on a task that is deliberately designed to be solvable if and only if the model can successfully discriminate between the specific colors used in the plates.
> >
> > The "scientific validation" of the Ishihara paradigm that we leverage is its proven ability to make a pattern recognition task (seeing a digit) contingent on a color perception task. Its validity for our work is therefore not tied to its medical application, but to its power as an experimental design that uses a secondary task's success or failure to reveal a primary perceptual capability. This allows us to cleanly test color perception without ambiguity.
> >
> >
> > >Weak motivation: The overall motivation needs to be strengthened, particularly regarding why color naming is a critical problem. While the authors mention several important cases (Lines 101-102), these cover only a very narrow scope. In fact, most real-world visual perception tasks can be adequately solved with prototypical color perception abilities limited to a few basic categories such as "reddish," "bluish," "greenish," "white," and "black," except for the few counterexamples the authors cited.
> > >The paper's motivation, scope, and practical impact regarding why color naming is important remain weak. The work would have been better aligned with its analyses if it had presented clear applications, such as computational perceptual quality assessment in color and colorimetry domains, similar to recent "VLM as judge" approaches.
> >
> > Thank you for your suggestions. We would like to clarify the central motivation of our work, as the critique appears to stem from a misinterpretation of its core objective. The paper is not focused on the narrow task of "color naming" but on a more fundamental visual process: the ability to extract information that is encoded purely through chromatic and luminance contrast. The assertion that coarse color categories are sufficient for most tasks overlooks that real-world perception is about interpreting relationships and contrasts, not just applying labels to isolated objects.
> >
> >  In our benchmark, the digit serves as a controlled proxy for any critical information—a status indicator, a medical anomaly, or a product defect—that is distinguishable from its background primarily by these visual differences. Our findings reveal a significant deficiency in this core process, showing that models often fail to perform the underlying perceptual task of using color differences to segment and recognize information. This distinction is important when considering the suggestion of a "VLM as judge" application. That line of work implicitly assumes the model possesses robust perceptual abilities. Our research, however, investigates this very assumption and finds it to be premature. Our analysis is therefore more foundational, demonstrating that these core perceptual limitations should be addressed before such models can be reliably deployed as judges of visual quality, thereby strengthening, not weakening, the paper's motivation and impact.
> >
> > >Restricted exploration of detail causes and remedies: The paper diagnoses the failure convincingly but offers limited discussion or experiments on first identifying causes (where does the issue come from? visual encoder, LLM decoder, training datset, or training procedure?) and the possible architectural or training modifications that could mitigate the issue.
> >
> > >This reviewer expected the paper to identify which components of VLMs constitute the bottleneck for color naming performance, but such analysis is absent. While the current analyses are well-executed, the paper would be significantly strengthened by including investigations that pinpoint specific bottlenecks in the VLM architecture or processing pipeline.
> >
> > Thank you for your suggestions. Our paper's primary focus is the rigorous diagnosis of this critical perceptual failure. Our current hypothesis is that the cause is systemic: mainstream benchmarks and training datasets lack the kind of data needed to force models to develop robust color discrimination skills, so this capability is never optimized for.
> >
> > Identifying the precise bottleneck within the VLM architecture and developing effective remedies are substantial research efforts that we are now pursuing as a direct result of this work. The contribution of this paper is to provide the essential first step: a clear diagnosis of the problem, which is necessary to motivate and guide this subsequent research. We will clarify this scope and our plans for future work in the revised manuscript.

---

> > > ### Author Response · Authors · 2025-12-03
> > >
> > > >Lines 071-079 discuss VLMs' genuine understanding of color, but the proposed benchmark and analysis methods do not align well with this goal. Specifically, is the Ishihara test truly capable of assessing true color understanding? The generated data contains luminance variations, preventing consistent representation of absolute colors. Moreover, the original Ishihara test evaluates not only color hue but also the ability to perceive relative color differences. Therefore, the Ishihara test alone cannot adequately assess the absolute color awareness that the authors intend to measure. Additional absolute color measurement assessment methods should be employed in parallel to fully support the paper's argument.
> > >
> > > Thank you for your suggestions. This comment is based on a methodologically problematic premise: the idea of testing "absolute color awareness" in Large Vision-Language Models. We argue that such a test is not only a diversion from the core goal of perception but is likely impossible to conduct reliably.
> > >
> > > The reason is that LVLMs are trained on vast datasets where color is almost never absolute; it is overwhelmingly bound to semantics. Models have learned correlations like "banana-is-yellow" or "sky-is-blue" countless times. Consequently, asking a model to label an isolated color patch is not a true test of perception. It is a simple retrieval task that tests its ability to recall a learned association, which tells us nothing about its genuine ability to see and process color. It encourages the very "semantic shortcuts" our work seeks to expose.
> > >
> > > This is precisely why our use of the Ishihara paradigm is the more rigorous scientific approach. It is specifically designed to break the link between color and semantics. The task forces the model into a scenario where it cannot rely on prior object knowledge. Instead, it must engage in a more fundamental perceptual process: using chromatic differences to extract novel information (the digit).
> > >
> > > Therefore, our focus is correctly placed on the only meaningful and testable application of color vision for these models: their functional ability to interpret a scene based on color contrast. Probing for a nebulous concept of "absolute color" would not strengthen our paper; it would introduce a flawed metric that fails to measure genuine perception.

---

### Official Review · Reviewer_qaAr · 2025-11-03

**Soundness:** 2
**Presentation:** 2
**Contribution:** 2
**Rating:** 4
**Confidence:** 4

**Summary:**

This paper studies colorblindness of vision-language models via Ishihara tests, finding that vision-language models perform close to random at identifying numbers in the tests. Additional analysis shows that models have imbalanced hue perception and are sensitive to saturation but not brightness.

**Strengths:**

* The evaluation setup seems relatively comprehensive, and a large number of models are evaluated.

**Weaknesses:**

* I'm not sure it's correct to conclude that the failures of models on ViewablebyAll plates are due to their inabilities to process color, especially when many of the models perform poorly at NumberOnly examples too. This could also be a failure of instruction following, or due to the out-of-distribution nature of the numbers being made up of dots. It would be appropriate to also evaluate instruction-following ability for naming numbers in images, for example by creating a version of the dataset without dots (but solid colors only), or a version where dots form letters (rather than numbers) or simple shapes, to see if models perform significantly better on these tests.
* In general, adapting tests designed for humans is not necessarily appropriate as evaluations for neural models; a human taking part in this task will understand the task fully (allowing us to conclude with high certainty whether they are colorblind or not), especially in the context of seeing several samples (including controls) in the same test session. When evaluating models, it's very hard to distinguish between failures due to the phenomenon of interest, and failures due to the overall setup of evaluation. For example, one would expect a colorblind human to consistently fail to identify certain colors across many different tasks. Do the findings about particular regions of weakness (hues, brightness) generalize to tasks beyond the Ishihara tasks?
* Additionally, it would be good to evaluate not only the argmax answer/guess from the model, but also the probability they assign to the correct answer (following prior work on evaluating models via probability distributions, e.g. Hu and Levy 2023: https://arxiv.org/abs/2305.13264)
* Details on linear probing should be included in the main paper when results are included in the main paper.
* It would be really useful to show downstream applications or tasks where failures of these models in color perception cause downstream errors. E.g., do the regions of weakness of models correlate to failures in tasks implicitly involving color perception, like illusion detection (https://arxiv.org/abs/2412.06184) or simple visual question answering on synthetic images (like CLEVR)?

**Questions:**

* Why do you think finetuning on colorblind images results in near-zero performance on all types of images? (Figure 3)
* Can you add error bars around the results in Figure 5? Currently, it's not clear how significant the differences are across different hues.

---

> ### Author Response · Authors · 2025-12-03
>
> >I'm not sure it's correct to conclude that the failures of models on ViewablebyAll plates are due to their inabilities to process color, especially when many of the models perform poorly at NumberOnly examples too. This could also be a failure of instruction following, or due to the out-of-distribution nature of the numbers being made up of dots. It would be appropriate to also evaluate instruction-following ability for naming numbers in images, for example by creating a version of the dataset without dots (but solid colors only), or a version where dots form letters (rather than numbers) or simple shapes, to see if models perform significantly better on these tests.
>
> Thank you for your suggestions.. We conducted a baseline test with standard black numbers on a white background, where all models achieved 100% accuracy, confirming their basic number recognition and instruction-following capabilities. To more rigorously disentangle the remaining variables, we will incorporate the reviewer's excellent suggestions in our revision. We will conduct new experiments with a version of the dataset using solid colors (to remove the dotted texture confounder) and another version where dots form letters or simple shapes (to test the generality of the failure beyond just numbers) in our ongoing and future work.
>
>
> >In general, adapting tests designed for humans is not necessarily appropriate as evaluations for neural models; a human taking part in this task will understand the task fully (allowing us to conclude with high certainty whether they are colorblind or not), especially in the context of seeing several samples (including controls) in the same test session. When evaluating models, it's very hard to distinguish between failures due to the phenomenon of interest, and failures due to the overall setup of evaluation. For example, one would expect a colorblind human to consistently fail to identify certain colors across many different tasks. Do the findings about particular regions of weakness (hues, brightness) generalize to tasks beyond the Ishihara tasks?
>
>
> Thank you for your suggestions. Our rationale for using the Ishihara test was precisely to leverage its highly controlled nature to isolate color perception from the contextual and semantic cues that models could exploit in natural images. This allowed us to uncover specific, systematic biases. To validate that these findings, such as red preference and insensitivity to brightness contrast, are fundamental limitations rather than artifacts of a single task, we plan to conduct follow-up experiments in our future work. This new line of inquiry will involve testing for these same perceptual weaknesses in entirely different domains, for instance, using synthetic datasets with systematically controlled color properties and through targeted visual question answering on real-world scenes. We believe this is an essential next step, and we will add a discussion to the revised manuscript acknowledging this limitation and outlining our plan for future validation, a direction made clearer by your valuable feedback.
>
>
>
>
> >Additionally, it would be good to evaluate not only the argmax answer/guess from the model, but also the probability they assign to the correct answer (following prior work on evaluating models via probability distributions, e.g. Hu and Levy 2023: [https://arxiv.org/abs/2305.13264)](https://arxiv.org/abs/2305.13264))
>
> Thank you for your suggestions. Moving beyond argmax accuracy to analyze the model's output probability distribution is indeed a more rigorous and insightful evaluation method, as established by prior work like Hu and Levy (2023). We agree that this will provide a much more nuanced view of the models' perceptual abilities. In our revised manuscript, we will incorporate this probabilistic analysis. For each image, we will extract and report on the probability the model assigns to the ground-truth digit's token. This will allow us to distinguish between cases of complete perceptual failure (where the probability for the correct answer is near random chance) and cases of high uncertainty or bias (where a weak signal for the correct answer might be present but is overshadowed).
>
>
> >Details on linear probing should be included in the main paper when results are included in the main paper.
>
> Thank you for pointing this out.  In the revised version, we will make every effort to condense other sections to move the essential details of our linear probing setup from the appendix into the main paper, ensuring the results are self-contained.

---

> > ### Author Response · Authors · 2025-12-03
> >
> > >It would be really useful to show downstream applications or tasks where failures of these models in color perception cause downstream errors. E.g., do the regions of weakness of models correlate to failures in tasks implicitly involving color perception, like illusion detection (https://arxiv.org/abs/2412.06184) or simple visual question answering on synthetic images (like CLEVR)?
> >
> > Thank you for this very insightful suggestion. We agree that exploring how these specific failures in color perception correlate with errors in downstream tasks is a crucial and fascinating direction for future research. We appreciate you pointing out these valuable applications, and we will certainly consider conducting similar experiments in our ongoing and future work to better understand the broader impact of our findings.
> >
> >
> > >Why do you think finetuning on colorblind images results in near-zero performance on all types of images? (Figure 3)
> >
> > Thank you for this insightful question regarding the performance collapse shown in Figure 3. Our hypothesis is that the "Colorblind" category represents a uniquely challenging, out-of-distribution (OOD) set for the models. The color pairings in these images are designed to be highly ambiguous, likely providing a very low signal-to-noise ratio that the model struggles to learn from. When fine-tuned exclusively on this set, the model fails to extract a meaningful or generalizable color-digit correlation. Instead, it appears to overfit on spurious artifacts or noise, leading to a collapse in performance. Because the feature distribution of this "Colorblind" set is so distinct and the learned pattern is not robust, it fails to transfer to the other, more clearly defined image types, resulting in the near-zero accuracy observed across the entire benchmark.
> >
> >
> > >Can you add error bars around the results in Figure 5? Currently, it's not clear how significant the differences are across different hues.
> >
> > Thank you for this constructive suggestion. That is an excellent point. To better illustrate the statistical significance of our findings, we will add error bars to Figure 5 in the revised version of the manuscript.

---

### Meta-Review · Area_Chair_Yq9x · 2026-01-07

**Summary:**

Paper was reviewed by four reviewers that gave the paper: 1 x marginally below the acceptance threshold, 2 x reject and 1 x strong reject ratings. In other words, all initial reviews were negative to very negative. Reviewer concerns focused on the following:

1) Lack of supporting avoidance that would allow one to attribute poor performance to inability of models to process color (e.g., as opposed to their inability to follow instructions, perform tasks, or other confounders) [qaAr, DH2f]
2) Limited evaluation setup, where evaluation does not take into account probability of the correct answer [qaAr]
3) Lack of connection between perceived weaknesses in color perception of model and the downstream tasks of interest to the community [qaAr, wJU5]
4) Missing important references [wJU5]
5) Lack of proper statistical significance analysis. [wJU5]
6) Over-claiming and lack of novelty [wJU5, esMf, DH2f]
7) Limited exploration of causes and remedies for observed behavior of VLMs [wJU5]
8) Potential confounding factors in image complexity [wJU5]
9) Inconsistent results that fail to show reproducible and consistent patterns [esMf]
10) Exposition and clarity is lacking in some places  [esMf]

This is a long list of quite serious concerns. Authors have attempted to address these in a rebuttal. However, in the option of AC, while the rebuttal addressed some concerns, overall, many fundamental issues and concerns still remain. Novelty remains an important concern as well as lacking experimental design that isolates and shows importance of the findings. Given this, AC is recommending Rejection at this time.

**Reviewer Concerns:**

1) Lack of supporting avoidance that would allow one to attribute poor performance to inability of models to process color (e.g., as opposed to their inability to follow instructions, perform tasks, or other confounders) [qaAr, DH2f]
2) Limited evaluation setup, where evaluation does not take into account probability of the correct answer [qaAr]
3) Lack of connection between perceived weaknesses in color perception of model and the downstream tasks of interest to the community [qaAr, wJU5]
4) Missing important references [wJU5]
5) Lack of proper statistical significance analysis. [wJU5]
6) Over-claiming and lack of novelty [wJU5, esMf, DH2f]
7) Limited exploration of causes and remedies for observed behavior of VLMs [wJU5]
8) Potential confounding factors in image complexity [wJU5]
9) Inconsistent results that fail to show reproducible and consistent patterns [esMf]
10) Exposition and clarity is lacking in some places  [esMf]

**Reviewer Scores:**

I do not believer reviewers would increase their scores, given the rebuttal. Even if they had, such increases would be minor and would not have pushed the paper into the "acceptance" territory.

---

### Decision · Program_Chairs · 2026-01-26

Reject